



# Using a process-based dendroclimatic proxy system model in a data assimilation framework: a test case in the Southern Hemisphere over the past centuries

Jeanne Rezsöhazy[1,2], Quentin Dalaiden[1], François Klein[1], Hugues Goosse[1], and Joël Guiot[2]

[1]Université catholique de Louvain (UCLouvain), Earth and Life Institute (ELI), Georges Lemaître Centre for Earth and Climate Research (TECLIM), Place Louis Pasteur, B-1348 Louvain-la-Neuve, Belgium
[2]Aix Marseille University, CNRS, IRD, INRA, College de France, CEREGE, Aix-en-Provence, France

**Correspondence:** J. Rezsöhazy (jeanne.rezsohazy@uclouvain.be)

**Abstract.** Currently available data assimilation-based reconstructions of past climate variations have only used statistical proxy system models to make the link between the climate model outputs and the indirect observations from tree rings. However, the linearity and stationarity assumptions of the statistical approach may have limitations. In this study, we incorporate the process-based dendroclimatic model MAIDEN into a data assimilation procedure, using as a test case the reconstruction of
near-surface air temperature, precipitation and winds in the mid-latitudes of the Southern Hemisphere over the past 400 years. We compare our results with a data assimilation approach including a linear regression as a proxy system model for tree-ring width proxies. Overall, when compared to instrumental data, the reconstructions using MAIDEN as a proxy system model offer a skill equivalent to the experiment using the regression model. However, knowing the advantages that a process-based model can bring and the improvements that can still be made with MAIDEN, those results are promising.

## 1 Introduction

Indirect observations of climate from proxies such as tree-ring width (hereafter TRW) or isotopic content in ice cores and coral inform on past climate variability beyond the instrumental era (Jones et al., 2009). Those proxy records have been used in numerous studies devoted to climate variability over the last millennium. This has been classically achieved by using empirical
(i.e., statistical) relationship between climate variables of interest (e.g., surface air temperature) and proxy observations (e.g. Mann et al., 1999; Luterbacher et al., 2004; Mann et al., 2008; Büntgen et al., 2011; Neukom et al., 2011; Tingley and Huybers, 2013; Wilson et al., 2016). Tree rings represent one of the main source of information for those studies as it is the most available and spatially distributed proxy to reconstruct past climates at high temporal resolution (e.g. Fritts, 1976; Jones et al., 2009; Cook et al., 1999; Esper et al., 2002; Wilson et al., 2016; Anchukaitis et al., 2017; Esper et al., 2018).



Although the network of TRW records is well developed, mostly in the extratropical latitudes of the Northern Hemisphere (PAGES 2k Consortium, 2017), these records only provide insight about past climates at a specific location. In order to spread the local information to the large spatial scale, statistical methods have been developed to provide spatially gridded climate reconstructions of the past centuries, generally on the basis of linear regressions between the network of proxy records and the climate field of interest, calibrated over the instrumental period (e.g. Fritts et al., 1971; Mann et al., 1998, 2007; Cook et al.,

2010; Wang et al., 2015; Anchukaitis et al., 2017). The use of linear regression is based on the assumption that the relationship between climate and tree growth is linear and stationary over time. However, many studies have highlighted the shortcomings associated with these assumptions for TRW records (Wilmking et al., 2020; Briffa et al., 1998; Wilson and Elling, 2004; Wilson et al., 2007; D'Arrigo et al., 2008; Guiot et al., 2014; Babst et al., 2018).

On the other hand, global climate model simulations offer a complete spatial picture of the past climate variability. Never-

theless, these simulations, if used alone, cannot follow the observed climate evolution because of the large impact of natural climate variability that is unpredictable over those timescales and thus cannot be reproduced by models, and because of potential biases. Over the last decades, data assimilation (DA) has been increasingly used in paleoclimatology to bring together the information from both indirect climate records from proxies and climate models to provide a more complete picture of the climate changes (Goosse et al., 2010; Widmann et al., 2010; Goosse et al., 2012; Steiger et al., 2014; Franke et al., 2017;

Steiger et al., 2018; Tardif et al., 2019).

Initially, the DA-based reconstructions were developed using temperature reconstructions derived from the statistical relationship between temperature and proxy observations, and not the proxy time series themselves (e.g. Crespin et al., 2009; Goosse et al., 2010). More recently, the use of forward proxy system models (hereafter, PSMs) has emerged in order to directly assimilate the proxy time series. Specifically, the PSMs make the link between the outputs of the climate model included in

the DA procedure and the assimilated proxy observations, for example TRW (Evans et al., 2013; Dee et al., 2016). To date, reconstructions based on data assimilation using observed tree-ring data have only included statistical PSMs (uni- or multivariate regressions; e.g. Franke et al., 2017; Steiger et al., 2018; Tardif et al., 2019). Contrary to statistical models, process-based dendroclimatic PSMs are able to account for the complexity of the relationship between climate and tree-ring proxy data by explicitly simulating the biological processes governing the climate dependency of tree-ring growth. In this regard, they may

overcome some of the limitations of statistical models. However, so far, they have never been used in a DA procedure with actual tree-ring proxy data, but only with pseudoproxies (Dee et al., 2016; Acevedo et al., 2017; Steiger and Smerdon, 2017). In these studies, the VS-Lite model (Tolwinski-Ward et al., 2011) has shown to significantly improve the quality of the reconstruction. However, VS-Lite cannot be considered as entirely process-based as it does not include any biological processes to drive tree growth, but is rather based on the implementation of the principle of limiting factors with threshold growth response

functions (Tolwinski-Ward et al., 2011).

In this study, we introduce for the first time a process-based dendroclimatic model in an offline data assimilation procedure with actual tree-ring width records. Among all the available mechanistic tree-growth models (e.g Misson, 2004; Dufrêne et al., 2005; Vaganov et al., 2006; Drew et al., 2010; Tolwinski-Ward et al., 2011), we chose to work with the ecophysiological model MAIDEN (Modelling and Analysis In DENdroecology; Misson, 2004). MAIDEN includes the atmospheric $CO_2$ concentration





among other inputs, which represents a major advantage for taking into account the potential effect of the recent exponential increase of $CO_2$ on tree growth (Myhre et al., 2013; Boucher et al., 2014). In Rezsöhazy et al. (2021), the MAIDEN model has been successfully applied to the PAGES2k TRW database using the calibration protocol developed in Rezsöhazy et al. (2020), highlighting the potential of MAIDEN to be used as a PSM for DA-based reconstruction of past climate variability.

    Here, we perform a comparative analysis on the impact of using a simple regression model or a process-based dendroclimatic

model as PSMs for TRW records in the DA framework. The goal is to evaluate if and how a complex process-based model of tree growth like MAIDEN can contribute to the improvement of large-scale DA-based reconstructions of past climate variability compared to the regression model classically used so far. Because MAIDEN has shown to be skillful in different regions of the Southern Hemisphere (SH; Rezsöhazy et al., 2021), we focus on that region and provide climate reconstruction of near-surface air temperature, precipitation and winds in the mid-to-high latitudes of the SH over the past 400 years. The

Southern Hemisphere also constitutes an interesting study case, as few reconstructions have been performed in this region so far (Neukom et al., 2011, 2014). The DA experiments are based on both TRW and ice core ($\delta^{18}$O and snow accumulation) proxy data to ensure the consistency of the reconstructed large-scale circulation pattern by avoiding to only reconstruct small-scale features. They both represent the best available continental annually resolved proxies in the SH (PAGES 2k Consortium, 2017).

First, DA experiments are performed and evaluated against state-of-the-art gridded climate datasets over the last century to evaluate the impact of the assimilation of the tree-ring data and the associated PSMs on the skill of the reconstruction. In Sect. 4.1, we compare the performance of the DA-based reconstructions using a regression- or a process-based dendroclimatic PSM in our framework over the last century. Sensitivity experiments are then performed in Sect. 4.2 to identify the contribution of tree-ring width proxies in our data assimilation framework using a regression- or a process-based dendroclimatic PSM as

well as to quantify the impact of the uncertainty of the records. Secondly, the DA-based reconstructions performed with a statistical or process-based dendroclimatic PSM are compared to each other over the past four centuries in Sect. 4.3. To match the seasonality of tree growth in the Southern Hemisphere, all the analysis in this study are performed on a July-June year.

## 2   Data assimilation

    In paleoclimatology, the objective of data assimilation is to optimally combine the information from the physics of the climate

system as included in climate models and indirect observations of climate from proxies (Kalnay, 2003; Goosse et al., 2010; Widmann et al., 2010; Steiger et al., 2014; Franke et al., 2017; Hakim et al., 2016). More specifically, the DA procedure is based on the Bayes theorem: starting from the prior estimate of the state of the climate system provided by the model, DA produces a reconstruction of the climate system as accurate as possible (i.e, the posterior) given the information provided by the climate observations. The updated estimate of the climate system is often called a reanalysis. One of the advantages of

DA is its ability to propagate the information brought by the observations both in space and between the climate variables, by relying on the spatial covariance of a specific variable and on the covariance between different variables in the model. In other



words, DA gives the opportunity to reconstruct the climate at places where there is no proxy sites, and for variables for which we do not have any observations to assimilate.

In this study, we use an offline DA approach, by opposition with the online DA approach. The offline approach samples a
pre-existing ensemble of climate model simulations to build the prior state of the climate system. This has the great advantage of being considerably computational time saving. In this configuration, the different years of the sampled ensemble are assumed to be independent of each other. This assumption is reasonable if the predictability of the variables of interest is smaller than the temporal resolution of DA (i.e, one year in our study), as is the case for atmospheric variables. With the online approach, the prior state of the climate system is sequentially updated. In this case, the years are not assumed to be independent of each
other. Instead, the update made at the previous time step is transferred to the next one. When focusing on the atmosphere, as in our study, the online approach is not expected to outperform the offline approach (Matsikaris et al., 2015; Okazaki et al., 2021).

Here, the offline approach is based on a particle filter method described in Sect. 2.1. In addition, the framework of DA includes three main components: (1) prior climate model simulations (Sect. 2.2); (2) indirect observations of climate provided by
proxy data (Sect. 2.3); and (3) proxy system models, to make the link between climate models outputs and indirect observations of climate (Sect. 2.4). The error associated with the data used in the DA procedure is also considered and defined in Sect. 2.5. Finally, in Sect. 3, we elaborate the DA experimental design and diagnostics of this study.

## 2.1   Particle filter

In this study, the offline DA procedure employs a particle filter as in Dubinkina et al. (2011). The particle filter (van Leeuwen,
2009) describes the prior of each climate variable by a probability density function, which is represented discretely on the basis of independent states of the model (Sect. 2.2), i.e., the particles. These model states are sampled at a chosen frequency in the model simulations, that determines the number of sampled particles. It will be characterized later in Sect. 3. The prior stays the same throughout the DA procedure. As the observations are annually resolved (see Sect. 2.3), the time step of the DA is also annual. As a consequence, the particles are defined as the annual mean of the model state variables.

For each year of the reconstruction period (i.e, the period covered by observations), all the particles are compared to the available observations through the use of a PSM (Sect. 2.4). This allows estimating the likelihood of each particle knowing the observations at that time. This computation also takes into account all known sources of errors (see Sect. 2.5). The likelihood then determines the weight given to each particle. The weight given to a particle will be higher if the particle is closer to the observations, and lower if further to the observations. In this way, starting from a prior distribution where all the particles
have the same weight, the particle filter produces a posterior distribution where the weights of the particles are redistributed according to the observations, for each year of the reconstruction. In this framework, the dynamical consistency of the model is preserved, since only the weights of the particles are updated. At the end of the DA procedure, the weighted mean of the particles for each year over the assimilation period provides an estimate of the climate variables (i.e., the ensemble mean). The weighted standard deviation of the particles is used to estimate the range of the reconstruction (i.e., the ensemble spread).



## 2.2 Climate model simulations

We use the ensemble of three simulations performed with the isotope-enabled Community Earth System Model version 1 (iCESM1; Brady et al., 2019; Stevenson et al., 2019; publicly available at https://www.earthsystemgrid.org/dataset/ucar.cgd. ccsm4.CESM_CAM5_LME.html; last access: 12 May 2021) as a basis for building the prior of our data assimilation experiments. The iCESM1 is a coupled atmosphere–ocean–sea-ice model at 1.9° latitude and 2.5° longitude resolution for the atmosphere component and around 1° for the ocean–sea-ice component. The model simulations include the anthropogenic (aerosol emissions and land use changes) and volcanic aerosol forcings, the greenhouse gas emissions, solar irradiance variations and Earth's orbital changes (Brady et al., 2019; Stevenson et al., 2019). The annually resolved iCESM1 simulations used in this study span the July 850 – June 2005 CE time period. For our DA experiments (Sect. 3), all the particles are sampled in the three iCESM1 members for a total of 3465 particles.

## 2.3 Proxy data

In this study, we make use of three types of proxy data from two types of archives: first, tree-ring width (Sect. 2.3.1) and, second, $\delta^{18}$O (i.e, the ratio of stable isotopes oxygen in the ice core) and snow accumulation from ice cores (Sect. 2.3.2). They represent the best available continental proxies in the SH continents (South America, Australia, Tasmania, New Zealand and Antarctica) for reconstructing the past climate at high-resolution (PAGES 2k Consortium, 2017).

### 2.3.1 Tree-ring width data

The PAGES2k database (hereafter PAGES2k2017) includes 692 proxy records (PAGES 2k Consortium, 2017) selected for reconstructing the temperature of the last two millennia. The records are from a variety of archives (tree rings, ice cores, lake sediments, coral, speleothems, documentary evidence, etc). TRW records represent a bit more than the half of the database with 354 records, among which 12 are located below 30°S, in New Zealand (four records), Tasmania (four records) and South America (four records). The previous version of the PAGES2k TRW database (PAGES 2k Consortium, 2013; hereafter PAGES2k2013) contains 28 TRW records below 30°S. Nine records have been excluded from the recent version of the PAGES2k database because they were too short (less than 300 years). Those records are also included in this study and are all located in South America. The location of the 21 TRW records from the two databases can be seen on Fig. S1 (blue stars). The location of the 15 TRW records that are actually used in this study can be observed on Fig. 1 (see Sect. 2.4.1 and 2.4.2 for more details on the included sites). The TRW records are normalized (i.e, with a null mean and a unity standard deviation) relative to the 1900–2000 CE time period for a use with the MAIDEN model (Sect. 2.4.1) while for the regression-based model (Sect. 2.4.2), the anomalies relative to the 1900–2000 CE time period are subtracted from the TRW time series.

### 2.3.2 Ice core data

The ice core data selected here are the same as in Dalaiden et al. (2021), a study focusing on the mechanisms behind the West Antarctic climate changes. This includes the 48 annually resolved ice core snow accumulation records in Antarctica





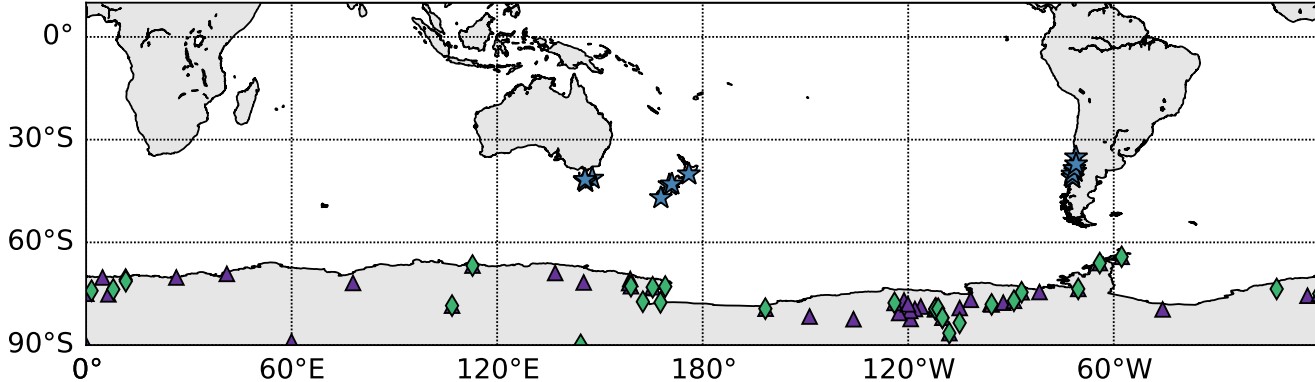

**Figure 1.** Location of the 15 tree-ring width (TRW) records (blue stars) from the PAGES2k databases (PAGES 2k Consortium, 2013, 2017) used in this study; the 49 ice core snow accumulation records (purple triangles) from Thomas et al. (2017) and Medley et al. (2018); and the 29 $\delta^{18}O$ records (green diamonds) from Stenni et al. (2017) (Sect. 2.3). Background map from Hunter (2007).

.

from the database of Thomas et al. (2017) spanning the last millennium (Medley and Thomas, 2019). The B40 ice core snow accumulation record of Medley et al. (2018) has also been added as this record has been published after the publication of the Thomas et al. (2017)'s database. Finally, as in Dalaiden et al. (2021), we use the 29 annually resolved $\delta^{18}O$ records in Antarctica from Stenni et al. (2017). Fig. 1 represents the location of the snow accumulation and $\delta^{18}O$ records (purple triangles and green diamonds, respectively). They are mostly situated in West Antarctica. The ice core records are averaged over a 500 km grid in order to reduce the non-climatic noise in the observations. This represents a maximum of 20 snow accumulation and 13 $\delta^{18}O$ time series over the last four centuries. The anomalies of the ice core data are computed relative to 1900–2000 CE.

## 2.4 Proxy System Models

Proxy System Models, also referred to as observation operators, are a crucial component of the DA framework that makes the link between the climate state variables of the prior such as near-surface air temperature and precipitation and the assimilated indirect observations from proxies, here tree-ring width. In this study, we make use of two types of tree-ring PSMs, at opposite ends of the PSM spectrum: first, a process-based ecophysiological model, MAIDEN (Sect. 2.4.1); second, a linear regression model (Sect. 2.4.2). If we neglect here potentially important elements such as post-depositional effects, the ice core records can be directly compared with the climate model. In particular, the difference between precipitation and sublimation/evaporation ($P - E$) from iCESM1 is compared to the snow accumulation observations. It has already been shown in different studies that the main contributor to the snow accumulation variability is indeed $P - E$, especially when working at large spatial scale (Agosta et al., 2019; Souverijns et al., 2018; van Wessem et al., 2018). Finally, the ratio of stable isotope oxygen $\delta^{18}O$ is simulated in iCESM1 and can be directly compared to $\delta^{18}O$ records.



### 2.4.1 MAIDEN

MAIDEN (Modelling and Analysis In DENdroecology; Misson, 2004; Gea-Izquierdo et al., 2015; Gennaretti et al., 2017) is a process-based dendroclimatic model which simulates the course of photosynthesis and the allocation of carbon to different pools (stem, roots, leaves, storage) on a daily time step as a response to climatic drivers (daily maximum and minimum temperature, precipitation and atmospheric $CO_2$ concentration). The annual quantity of carbon which is allocated to the stem (hereafter Dstem) is assumed to be proportional to tree-ring growth and comparable to TRW data, after normalization. Therefore, this variable is used for the comparison with the observed TRW. We use a combined version of the model from Gea-Izquierdo et al. (2015) and Gennaretti et al. (2017), developed by Fabio Gennaretti (unpublished).

We must first determine where the MAIDEN model can be skillfully applied among the 21 available PAGES2k sites (Sect. 2.3.1). MAIDEN was calibrated at the 12 SH sites from PAGES2k2017 in Rezsöhazy et al. (2021) driven by the daily maximum and minimum temperature and precipitation from the *Global Meteorological Forcing Dataset for land surface modelling* (v2) (http://hydrology.princeton.edu/data.php, last access: 5 May 2021; Sheffield et al., 2006) at 0.5° resolution (hereafter GMF), over 1950–2000 CE. The model was validated using the same dataset over the 1901–1949 CE time period. The calibration method is based on a Bayesian procedure with Markov Chain Monte Carlo sampling using the DREAMzs algorithm (Hartig et al., 2019). This Bayesian approach is used to calibrate the most sensitive parameters of the model from the photosynthesis and carbon allocation modules (18 parameters). MAIDEN was found to be robustly calibrated with significant (95% confidence level) calibration and verification correlations greater than 0.3 at five sites south of 30°S. The same methodology was used here for calibrating and validating MAIDEN at the nine records of PAGES2k2013. One site was found to be well calibrated and validated. The six sites for which MAIDEN calibration has successfully passed those tests will thus be incorporated into the DA procedure with MAIDEN. They can be seen on Fig. S2 (orange circles). In particular, there is no site in Tasmania.

The iCESM1 data used in the DA framework are available at a monthly resolution. Yet, MAIDEN runs on a daily basis. To convert monthly data into daily data, we apply the same methodology to all the particles. We use a reference year of the daily GMF dataset (1950) which we modified to match the mean seasonal cycle over 1950–2000 CE, i.e., a climatological seasonal cycle. Then, for a given particle, the monthly anomalies of the iCESM1 data relative to 1950–2000 CE are added (near-surface air temperature) or multiplied (precipitation) to the reference year of the daily GMF data. Note that the iCESM1 does not include maximum and minimum temperature data. We use the anomalies of iCESM1 near-surface air temperature on both maximum and minimum daily GMF temperature. As for atmospheric $CO_2$ data, the annual data from Rubino et al. (2019) (154–1996 CE; publicly available at https://doi.org/10.25919/5bfe29ff807fb, last access: 10 January 2021) followed from 1996 by the annual data from Sato and Schmidt (1880–2011 CE; https://data.giss.nasa.gov/modelforce/ghgases/, last access: 10 January 2021) were linearly interpolated at a daily time step.

The temporal autocorrelation in the simulated Dstem can be important, through the storage pool (Gea-Izquierdo et al., 2015; Gennaretti et al., 2017; Guiot et al., 2014), so that the performance of the model in reproducing the observed tree growth of one year highly depends on the performance of the previous years. As a consequence, we cannot run the MAIDEN model independently for each year driven by the iCESM1 data as is the case with the linear regression (Sect. 2.4.2). Instead, MAIDEN





is sequentially run year by year using the reconstructed climate for the previous years and thus accounting for the effect of past
climate history on tree-ring growth. For any particle, we use the $CO_2$ data corresponding to the assimilated year.

Dstem is not directly comparable with TRW data. The observations and simulation results are thus normalized (Sect. 2.3.1)
to produce unitless indexes. The mean and standard deviation of a reference run of MAIDEN with iCESM1 inputs over 1900–
2000 CE are used to normalize the annual Dstem output throughout the DA procedure.

### 2.4.2 Linear regression

A regression-based model was developed for each of the 21 TRW records (Sect. 2.3.1), similarly to Dalaiden et al. (2021) and
to the Last Millennium Reanalysis (Tardif et al., 2019). Different climate variables were considered to model the relationship
between climate and tree-ring growth. They are either related to temperature, precipitation, or both (univariate or bivariate
model). For temperature, we considered the current year July to next June temperature, the December-January-February (i.e.,
austral summer) temperature and the September to March (i.e., the growth period temperature). For precipitation, we considered
the July to June precipitation, the July-August precipitation and the July-August-September precipitation. We built the different
climate variables by averaging the monthly near-surface air temperature and precipitation from the GMF interpolated on the
iCESM1 grid (Sect. 2.2) over the period of months of interest. The climate data were extracted from the continental grid cell
closest to each individual TRW record.

Similarly to the MAIDEN calibration, the regression parameters of each statistical PSM were first calibrated over the 1950–
2000 CE time period using the ordinary least squares method. For each record, we tested all the possible combinations of
climatic variables. As in Dalaiden et al. (2021), the best model was selected for each TRW record using the *Bayesian Infor-
mation Criterion* (BIC) (Schwarz, 1978), which has the advantage to penalize the complexity (i.e, the number of explanatory
variables) of a model.

The regression models were then run independently over the 1901–1949 CE verification time period. Among the 21 sites,
only two sites were validated following the same criterion as MAIDEN (Sect. 2.4.1), i.e, a significant calibration and verifi-
cation correlations greater than 0.3 (Table S1). As a consequence, the validity of the regression models out of the calibration
interval is not ensured for most of them. Alternatively, we decided to use a less strict selection criterion similar to what is
generally done in paleoclimate DA-based reconstruction using TRW proxies with a statistical model (e.g., the Last Millennium
Reanalysis; Tardif et al., 2019). The regression parameters are thus calibrated using the same method over the entire period
covered by climate data, i.e., 1901–2000 CE, in order to have the largest possible sample and thus the most robust estimate of
the relationship between climate and tree growth. We then excluded nine sites (one in New Zealand and eight in South Amer-
ica) for which the explanatory climate variables were not significantly and positively correlated with observed TRW series. The
12 sites included in the DA procedure with the regression model (hereafter, selected sites) can be seen on Fig. S3 (blue circles).
A similar criterion based on the calibration only could not be used to select the TRW records for MAIDEN. The complexity
of the model and the number of calibrated parameters make it more prone to overfitting problems and biological inconsistency
during the calibration (Rezsöhazy et al., 2020, 2021). As a consequence, the selection of the TRW records has to be more





restrictive but may also lead to a better stability of the model. For comparison with the observed TRW series, anomalies of the simulated tree-ring indexes are computed relative to 1900–2000 CE.

## 2.5 Observation error

In the DA framework, the observation error determines the confidence that we can have in the climate information retrieved from the available proxy records. It directly influences the magnitude of the weight given to a particle when compared to an assimilated proxy observation, which has in turn direct consequences on the resulting reanalysis. We generally identify three types of observation error (e.g. Badgeley et al., 2020).

The first type of error is related to the inaccuracy of the PSM in reproducing the proxy dependency on climate as the PSMs
cannot fully account for the complexity of the processes by which tree growth responds to climate and non-climatic drivers. The error of MAIDEN is computed using the standard deviation of the residuals over the verification period (1901–1949 CE; Sect. 2.4.1). The error of the TRW regression-based PSM (Sect. 2.4.2) is computed as in Tardif et al. (2019), using the standard deviation of the residuals over the calibration period (1901-2000 CE). However, computing the error over the calibration period can have serious shortcomings. It does not account for the actual skill of the model to accurately estimate tree growth outside
the calibration interval and is biased by possible overfitting problems associated with any statistical model. This could result in an underestimation of the error associated with the statistical PSM and, as a consequence, to overconfidence in the PSM performance and finally in the climate reconstruction. The snow accumulation and $\delta^{18}$O records can be directly compared to the model outputs (Sect. 2.4) so that a PSM is not necessary and therefore an error related to the performance of the PSM is not needed for these records.

Second, the representativeness error (e.g. Oke and Sakov, 2008) corresponds to the uncertainty related to the unresolved processes at the resolution of the climate model relative to the smaller scale potentially represented by proxy observations (tree-ring width, snow accumulation and $\delta^{18}$O in our study). For the regression-based TRW PSM, we consider that the representativeness error is implicitly taken into account by computing the error related to the PSM using observed data at the same resolution as the iCESM1 data (Sect. 2.4.2). For MAIDEN, the bias correction applied to the iCESM1 data compared with the
high-resolution GMF data (Sect. 2.4.1) used for the calibration of the model is associated with an error but we assume that it is weaker compared to the error related to the PSM.

Regarding the ice core records (snow accumulation and $\delta^{18}$O), the representativeness error was computed as in Dalaiden et al. (2021), on the basis of an Antarctic simulation performed with the latest version of the polar-oriented Regional Atmospheric Climate MOdel (RACMO2.3p2, hereafter RACMO) at 27 km horizontal resolution (van Wessem et al., 2018). They
computed the standard deviation of the differences between the annual RACMO time series (1979–2016 CE) of snow accumulation averaged on all the ice core sites and the time series averaged over the 500 km grid, following the same approach as in Valler et al. (2020). As for the $\delta^{18}$O, the same methodology was applied but with the iCESM1 linearly interpolated on the RACMO grid over 1950–2005 CE, as RACMO does not simulate the water isotopes. The error was then corrected to account for the underestimation of spatial variability of snow accumulation and $\delta^{18}$O in models compared to point measurements as
ice cores, by multiplying the resulting standard deviation by a factor three (see Dalaiden et al. (2021) for more detail). The



last type of error is the instrumental error related to the measurement of the proxy data, often negligible compared to the other sources of error, particularly in paleoclimatology (Steiger et al., 2018; Tardif et al., 2019).

## 3 Experiment design and diagnostics

Different DA experiments are run in this study (Table 1), varying between them by (1) the records assimilated, (2) the tree-ring
PSM included and its associated period of calibration, (3) the PSM error and (4) the period of the reconstruction.

A first set of experiments is conducted over the 1900–2000 CE time period and compared with gridded climate datasets in order to evaluate the performance of our DA-based reconstructions regarding the use of statistical (TIC-reg) and process-based (TIC-MDN) dendroclimatic models separately, assimilating both ice core and tree-ring proxies.

A second set of experiments is performed over the 1900-2000 CE time period to evaluate the contribution of tree-ring
proxies in the data assimilation framework over the last century. Specifically, we perform an experiment assimilating tree-ring proxies only with the regression (TREES-reg) and MAIDEN (TREES-MDN) models and an experiment assimilating ice core proxies only (IC). These experiments are compared with the TIC-reg and TIC-MDN experiments incorporating both types of proxy data. The impact of the observation error associated with the tree-growth PSMs (Sect. 2.5) on the performance of the reconstruction is also evaluated by reducing the error by a factor two in the experiments constrained by tree-ring proxies only
(TREES-reg-05 and TREES-MDN-05).

Lastly, we perform a DA-based reconstruction over the period covered by a sufficient number of TRW proxy records, i.e., 1600–2000 CE, with the regression model (TIC-reg) and the MAIDEN model (TIC-MDN), with both tree-ring and ice core proxies assimilated.

Our analysis concentrates on four regions or sub-regions of the mid-latitudes of the SH: South America, defined between
25° and 56°S and between 55° and 85°W (SA); the South-Western Andes, where the TRW proxy sites are located, defined between 35° and 42.5°S and between 70° and 85°W (SAmSW); New Zealand, defined between 29° and 51°S and between 160° and 179°E (NZ); and Tasmania, defined between 39.5° and 47°S and between 140° and 154°E (Tas). We did not include Antarctica in our study as tree-ring records have not shown to significantly affect the reconstruction in this region (see Dalaiden et al. (2021) for more details on the reconstructions in Antarctica).

We evaluate our reconstructions against the most recent atmospheric reanalysis from the ECMWF, ERA5 (Hersbach et al., 2020) over the 1979–2000 CE time period, and against the GMF dataset (see Sect. 2.4.1 for more details on the dataset) over 1901–2000 CE, after linearly interpolating them on the iCESM1 grid (Sect. 2.2). We focus on the near-surface air temperature, the cumulative precipitation and the 500-hPa geopotential height (ERA5 only). In addition to these variables, we also evaluate the Southern Annular Mode (SAM) over 1957–2000 CE. The SAM is the principal mode of atmospheric variability in the
extratropics of the SH (Marshall, 2003). It is associated with the north-south movement of the Westerlies and often defined as the difference between the normalized monthly zonal mean sea-level pressure at 40° and 65°S (Gong and Wang, 1999; Marshall, 2003), as chosen here. The SAM index from Marshall (2003) is used as a reference for our evaluation.



**Table 1.** Data assimilation experiments in this study and experimental design: experiment name; types of proxies assimilated; Proxy System Model (PSM) used to assimilate the proxy data; error associated with the PSM; calibration period of the PSM; the proxy records assimilated. *N.A* stands for Not Applicable. See Sect. 2.4.1 and 2.4.2 for details on the *Validated* and *Selected* sites, respectively. The number of tree-ring width records assimilated is in brackets.

| Experiment name | Proxies | PSM | PSM error | Calibration period of PSM | Proxy records |
|---|---|---|---|---|---|
| **TIC-reg** | Ice cores | N.A | N.A | N.A | All |
| | Tree-ring width | Regression-based | Calibration | 1901–2000 CE | Selected (12) |
| **TIC-MDN** | Ice cores | N.A | N.A | N.A | All |
| | Tree-ring width | MAIDEN | Validation | 1950–2000 CE | Validated (6) |
| **TREES-reg** | Tree-ring width | Regression-based | Calibration | 1901–2000 CE | Selected (12) |
| **TREES-MDN** | Tree-ring width | MAIDEN | Validation | 1950–2000 CE | Validated (6) |
| **IC** | Ice cores | N.A | N.A | N.A | All |
| **TREES-reg-05** | Tree-ring width | Regression-based | Calibration error divided by two | 1901–2000 CE | Selected (12) |
| **TREES-MDN-05** | Tree-ring width | MAIDEN | Validation error divided by two | 1950–2000 CE | Validated (6) |

For each of the experiments, the performance of our reconstruction is evaluated with the computation of two statistical indicators: the Pearson correlation coefficient ($r$; considered as significant at the 95% confidence level) and its associated p-value; and the coefficient of efficiency (hereafter CE). The coefficient of efficiency is computed as in Nash and Sutcliffe (1970):

$$CE = 1 - \sum_{i=1}^{n} \frac{(x_i - y_i)^2}{(x_i - \bar{x})^2} \tag{1}$$

with $n$, the number of samples; $x$, the observation vector; $y$, the reconstruction vector. The overbar characterizes a time-averaged vector of values. The average is made over 1901-2000 CE in all experiments, except for the SAM index when compared to the observations from Marshall (2003), as they only cover the 1957-2000 CE period. While the Pearson correlation coefficient gives an estimation of the linear fit between the reconstructed and observed values, the CE also informs on the amplitude of the observed and reconstructed signal. For regional near-surface air temperature and precipitation, the metrics are computed between the mean of the reconstruction and the mean of the ERA5 or GMF data over the continental mass that lies in the region delimited by the latitudes and longitudes above.





## 4   Results and discussion

### 4.1   Comparison of the reconstructions using regression- and process-based dendroclimatic proxy system models over the last century

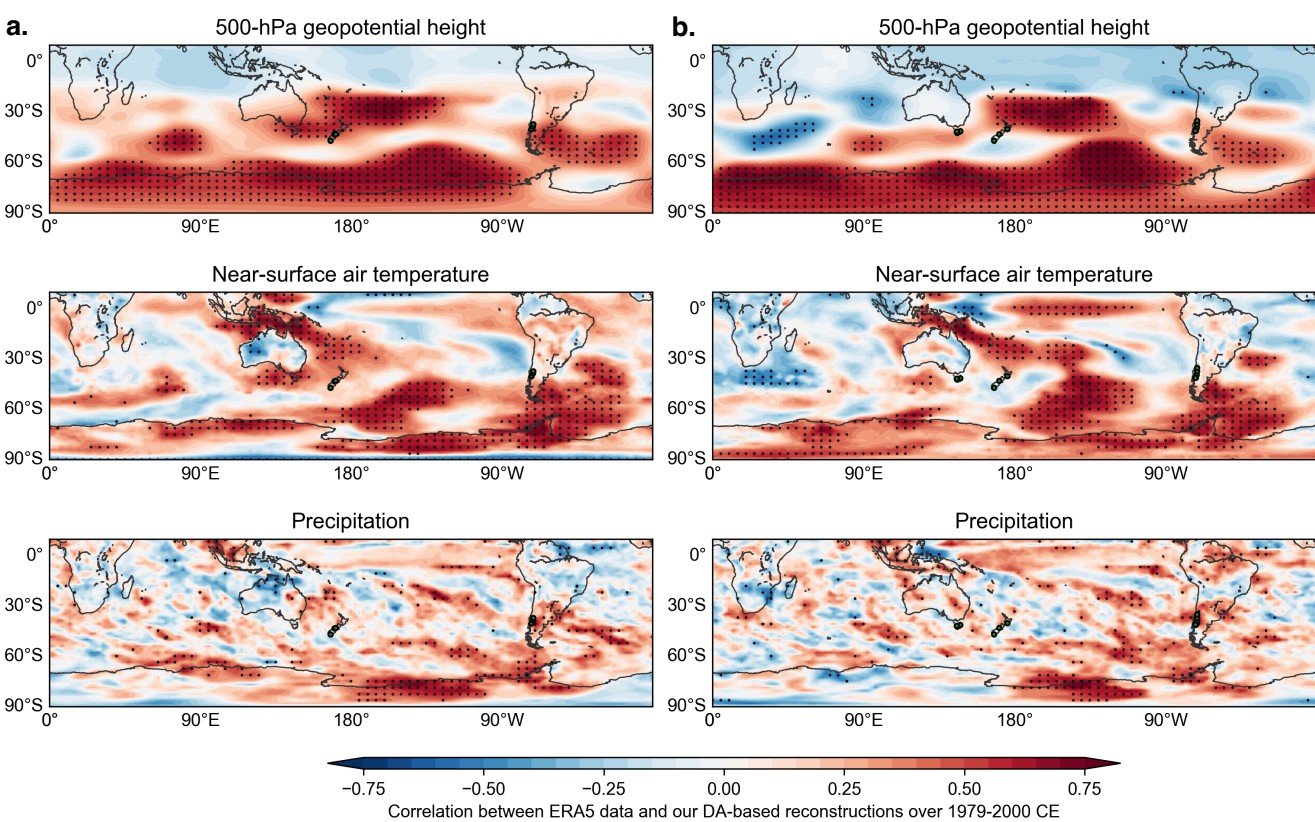

**Figure 2.** Pearson correlation coefficient between the reconstructed and ERA5 geopotential height at 500 hPa, near-surface air temperature or cumulative precipitation over 1979–2000 CE for the TIC-MDN (a) and TIC-reg (b) experiments (see Sect. 3 and Table 1 for details on the experiments). The green dots indicate the localization of the assimilated tree-ring width records. The black dots indicate a significant correlation coefficient at the 95% confidence level.

The spatial distribution of the correlation coefficients between different ERA5 and reconstructed climate fields over the recent period (1979–2000 CE) indicates a general good agreement of the reconstruction using MAIDEN as a PSM south of 30°S (TIC-MDN; Fig. 2a). When compared with the DA experiment including the regression model to assimilate 12 TRW records (TIC-reg; Fig. 2b), the hemispheric 500 hPa geopotential height, near-surface air temperature and precipitation fields are reconstructed with a similar performance in both cases.

At the regional scale and over the recent period (1979–2000 CE), the correlation coefficients of the ERA5 and GMF near-surface air temperature with the reconstructed near-surface air temperature in the four regions (Table 2) are also generally



similar for both TIC-reg and TIC-MDN. Except in Tasmania (Tas), where the TIC-MDN near-surface air temperature is highly
and significantly correlated with both ERA5 and GMF, the correlation coefficients are low and not statistically significant for
both experiments. This differs from the good general pattern previously noted (Fig. 2). Locally, the correlations can indeed be
lower (see Figs. S4–S9 for the regional pattern of the correlations). It is also important to highlight that these correlations have
been computed between the integrated fields and are not the average of the correlations represented on the map, which may

result in slight differences. Regarding precipitation (Table 2), the correlation coefficients are particularly high and significant
in South America (SA) and in the South-Western Andes (SAmSW) for TIC-reg and TIC-MDN.

**Table 2.** Pearson correlation coefficient between the reconstructed and GMF and ERA5 near-surface air temperature or precipitation over
1979–2000 CE for different DA experiments with both ice core and tree-ring proxies (Table 1) and for different regions (Sect. 3). *SA* stands
for South America; *SAmSW* for the South-Western border of South America in the Andes where the TRW proxy sites are located; *NZ* for
New Zealand; *Tas* for Tasmania. Asterisks stand for significant correlation coefficients at the 95% confidence level.

|  | ERA5 | | GMF | |
| --- | --- | --- | --- | --- |
|  | TIC-reg | TIC-MDN | TIC-reg | TIC-MDN |
| **Near-surface air temperature** | | | | |
| SA | -0.07 | 0.04 | 0.12 | 0.20 |
| SAmSW | -0.15 | 0.13 | -0.21 | 0.14 |
| NZ | 0.11 | 0.33 | 0.14 | 0.32 |
| Tas | 0.05 | 0.56* | 0.17 | 0.52* |
| **Precipitation** | | | | |
| SA | 0.72* | 0.71* | 0.65* | 0.64* |
| SAmSW | 0.54* | 0.44* | 0.53* | 0.44* |
| NZ | 0.25 | 0.26 | 0.28 | 0.44* |
| Tas | 0.10 | 0.00 | 0.16 | 0.14 |

Over the whole century (1901–2000 CE), both DA experiments have similar skills (ranging from -0.12 to 0.60 for the
correlation coefficients, and from -0.12 to 0.36 for the CE, for the four regions; Fig. 3; Table 3) in reconstructing the near-
surface air temperature when compared to the GMF climate dataset. Except in the South-Western Andes region (SAmSW in

Table 3), the correlation coefficients and CE between reconstructed and GMF temperature are close for both experiments and
particularly high in New Zealand (NZ). In Tasmania, no TRW records have been assimilated with MAIDEN, which may explain
the lower CE (Table 3) for the TIC-MDN experiment, while the correlation coefficients are similar between both experiments.
As for precipitation (Table 3 and Fig. 4), both TIC-MDN and TIC-reg are more skillful in the vicinity of the South American
TRW proxies (SAmSW), with significant correlation coefficients of 0.35 and 0.37, respectively, and positive CE.

Overall, the variance of the ensemble mean reconstruction is far smaller than the observed variance, for both near-surface
temperature and precipitation (Figs. 3 and 4), but observations are generally well in the range of the reconstruction (except for
extreme years). Since this pattern is apparent in both experiments, i.e., when using a regression and when using a process-based



**Table 3.** Pearson correlation coefficient and coefficient of efficiency (CE) between the reconstructed and GMF near-surface air temperature or precipitation over 1901–2000 CE for different DA experiments with both ice core and tree-ring proxies (Table 1) and for different regions (Sect. 3). *SA* stands for South America; *SAmSW* for the South-Western border of South America in the Andes where the TRW proxy sites are located; *NZ* for New Zealand; *Tas* for Tasmania. Asterisks stand for significant correlation coefficients at the 95% confidence level.

| | Correlation | | CE | |
|---|---|---|---|---|
| | TIC-reg | TIC-MDN | TIC-reg | TIC-MDN |
| **Near-surface air temperature** | | | | |
| SA | 0.07 | -0.12 | -0.06 | -0.12 |
| SAmSW | 0.34* | 0.00 | 0.12 | -0.07 |
| NZ | 0.55* | 0.54* | 0.30 | 0.27 |
| Tas | 0.60* | 0.52* | 0.36 | 0.18 |
| **Precipitation** | | | | |
| SA | 0.12 | -0.05 | 0.00 | -0.06 |
| SAmSW | 0.37* | 0.35* | 0.12 | 0.12 |
| NZ | -0.01 | 0.10 | -0.12 | -0.02 |
| Tas | 0.08 | 0.04 | -0.02 | -0.02 |

model as a PSM, this is likely not predominantly related to the intrinsic characteristics of each PSM. Our DA experiments only include 6 and 12 TRW proxy sites in the mid-latitudes of the SH for TIC-MDN and TIC-reg, respectively. Comparatively,

in the high latitudes, the ice core records are far more numerous (Sect. 2.3.2). However, the ice core records can only have an influence on the reconstructed variance over a limited area. As a consequence, the number of assimilated TRW proxies in the mid-latitudes is probably not sufficient to effectively constrain the climate model simulations during the DA procedure. In the DA framework, when no proxy is assimilated, the posterior as defined in Sect. 2.1 is equivalent to the prior, in which all the particles have the same weight. The posterior and the resulting ensemble mean are thus the same for each year of the

reconstruction. As a consequence, the reconstruction has, by construction, a variance equal to zero. For a small number of proxies providing a weak constrain, many particles receive a similar weight, resulting in a small variance too. By contrast, the uncertainty of the reconstruction (as measured by the standard deviation of the posterior or the ensemble spread; Sect. 2.1) remains large. Additionally, both the MAIDEN and regression models work on the basis of normalized observed and simulated tree-ring growth indexes. Consequently, in the DA procedure, the error in the simulated variance of the climate signal is likely

not properly assessed with the TRW PSMs. It is in practice possible to transform tree-ring data in order to directly compare them with the output of the MAIDEN model, without normalization (Gennaretti et al., 2018). This implies to combine both tree-ring width and density measurements, that are less widely available (PAGES 2k Consortium, 2017). This could improve the model-data comparison in the DA procedure, and, consequently, the resulting reconstruction. This is not the case for the regression model.





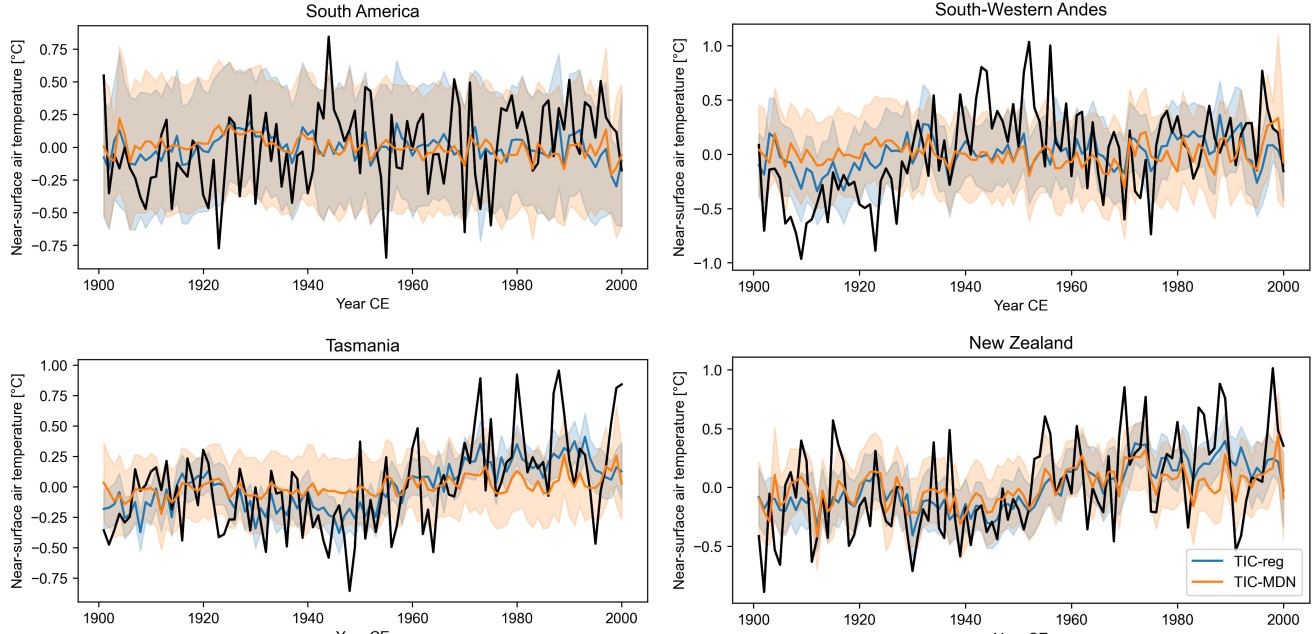

**Figure 3.** Reconstructed (in color) and GMF (in black) near-surface air temperature anomalies (in °C; relative to 1901–2000 CE) over 1901–2000 CE for different DA experiments with both ice core and tree-ring proxies (Table 1) in different regions (Sect. 3). The uncertainty associated with each reconstruction is represented by the shaded areas.

The DA procedure seeks to identify the climate conditions (i.e., the inputs of the TRW PSM; near-surface air temperature and precipitation) that gave rise to the observed TRW, through the use of a PSM. As a consequence, to fully verify our DA procedure, we must also establish if the reconstruction of near-surface air temperature and precipitation translates into a local ability of the MAIDEN model to reproduce tree growth at the six assimilated TRW sites over the 20th century. First, we have retrieved the monthly climate fields from the ensemble mean of the TIC-MDN reconstruction. Then, the monthly climate inputs

have been converted to daily inputs as in Sect. 2.4.1. For consistency, the tree-growth time series of the MAIDEN model have been normalized with the mean and the standard deviation of the reference run used in the DA procedure (Sect. 2.4.1). If we only look at the correlation coefficients between the observed and simulated tree-growth indexes (Fig. S10), MAIDEN has been successfully run over 1900–2000 CE with the TIC-MDN reanalysis of near-surface air temperature and precipitation as inputs at most assimilated sites (with a mean correlation of 0.59). The TRW sites displaying lower correlation coefficients between

the observed and simulated tree-ring indexes are the ones associated with higher PSM errors (Sect. 2.5), i.e., Aus_002 and SAm_024. This result is logical as a lower confidence is given to the latter in the particle filter (Sect. 2.1), so that the climate reconstructed at the local scale is less influenced by those records. However, the simulated variance is strongly underestimated compared to the observations, as seen on Fig. S10. This underestimation of the variance of tree growth directly comes from the





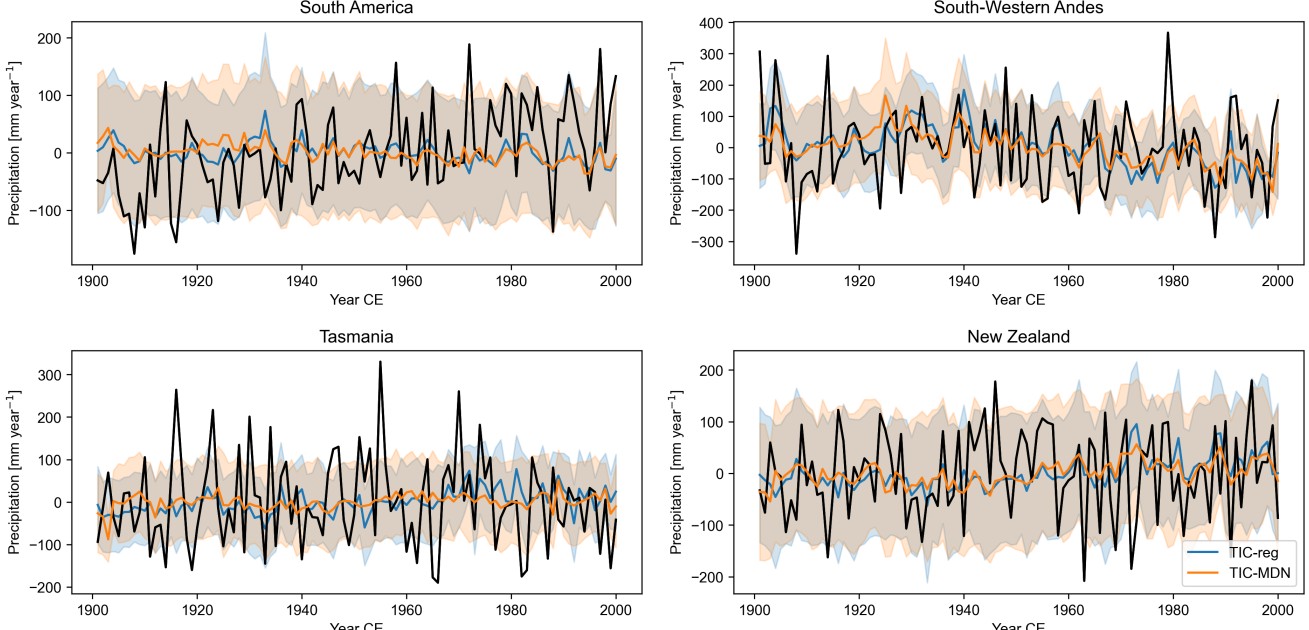

**Figure 4.** Reconstructed (in color) and GMF (in black) precipitation anomalies (in mm year$^{-1}$; relative to 1901–2000 CE) over 1901–2000 CE for different DA experiments with both ice core and tree-ring proxies (Table 1) in different regions (Sect. 3). The uncertainty associated with each reconstruction is represented by the shaded areas.

underestimation of the variance in the reconstructed near-surface air temperature and precipitation, as highlighted above. The

same pattern of small simulated variance is observed with the tree-ring indexes simulated with the linear regression.

Finally, the reconstructed SAM index is well and significantly correlated with the SAM index from Marshall (2003) for TIC-MDN (0.46) and TIC-reg (0.50) over the 1957–2000 CE period (Fig. 5). However, the TIC-reg experiment slightly better captures the significant observed trend of 0.28 standard deviation per decade over 1957-2000 CE, with a significant reconstructed trend of 0.34 and non-significant reconstructed trend of 0.20 standard deviation per decade for TIC-reg and TIC-MDN, respec-

tively. However, both reconstructions display negative values for the CE, which indicates that they are unable to accurately reproduce the observed amplitude of the SAM variations.

Overall, the DA experiment using MAIDEN as a PSM for assimilating the TRW records does not outperform the experiment using the regression model. However, its skill is always as good as the TIC-reg experiment with a potential better stability of MAIDEN through time, since it has been independently validated at the assimilated sites. It is also worth noting that the DA

experiments with MAIDEN or the regression model have some characteristics that can influence their respective skills. Firstly, the DA experiment with the regression model includes twice the number of TRW records compared to the experiment with MAIDEN (12 and six, respectively). The TRW records can also be located at different places which may result in different local skills of the DA experiments (e.g., in Tasmania). Secondly, the regression model was calibrated over the whole century





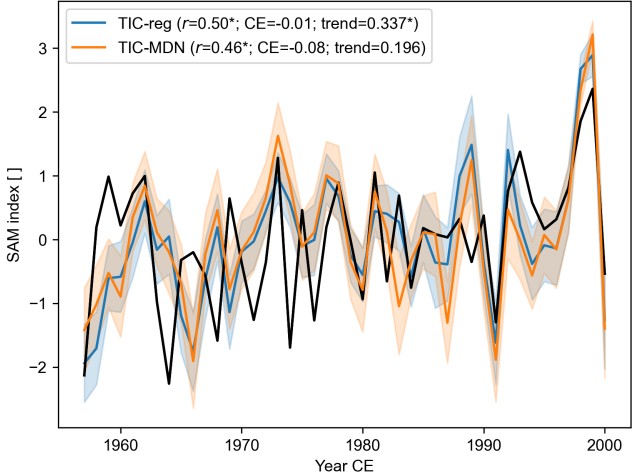

**Figure 5.** Z-scored (relative to 1957-2000 CE) Southern Annular mode (SAM) index of Marshall (2003) (in black) and reconstructed SAM index (in color) over 1957–2000 CE for different DA experiments with both ice core and tree-ring proxies (Table 1); Pearson correlation coefficient ($r$), coefficient of efficiency (CE), and decadal trend of the reconstructed SAM index (trend). Asterisks stand for a significant correlation coefficient or trend at the 95% confidence level. The observed trend of the SAM index is 0.28* standard deviation per decade over 1957–2000 CE. The uncertainty associated with each reconstruction is represented by the shaded areas.

(1901–2000 CE), while MAIDEN was calibrated over the second half of the century. This may confer a comparative advantage

to the regression-based DA experiment, particularly when evaluated against the GMF climate dataset over 1901–2000 CE.

### 4.2 Contribution of tree-ring proxies to the skill of the reconstructions over the last century

The experiments discussed in Sect. 4.1 include different types of proxies. Evaluating the specific contribution of tree-ring proxies using a regression- or process-based PSM, we focus on the regions and climate variables for which tree-ring data could likely provide a strong constrain (Sect. 4.1). This evaluation is essential to ensure that the few tree-ring proxies assimilated

– compared to the ice core data – are efficiently taken into account in a realistic framework including proxies in different regions using both MAIDEN or the regression-based PSM, and if differences arise between the experiments with these PSMs. Complementary information can be obtained by analyzing experiments in which only a fraction of the proxies is assimilated.

In South America (SA; Table 4 and Fig. 6), the correlations with observations are low for the experiments with tree-ring proxies alone, whatever the tree-ring PSM used (TREES-reg and TREES-MDN). Ice core proxies alone (IC) perform better

than the tree-ring proxies alone (TREES-reg and TREES-MDN) or when combined with the tree-ring proxies (TIC-reg and TIC-MDN) in terms of correlation coefficients, but not in terms of reconstructed variance (standard deviation for TIC-reg, TIC-MDN, TREES-reg or TREES-MDN higher than IC). However, as previously noted, it is clear that, even though the reconstructions with tree-ring proxies display a higher variance, none of the DA experiments accurately reconstruct the observed variance of precipitation (Fig. 6).





At the more local scale, in the South-Western Andes (SAmSW), the tree-ring proxies alone provide a better reconstruction
skill than for the South America (TREES-reg and TREES-MDN; Table 4). The correlation coefficients are, however, slightly
lower than in the experiments using both tree-rings and ice core proxies or ice core proxies alone, in particular for TREES-
MDN. Nevertheless, the TREES-reg and TREES-MDN reconstructions have a higher reconstructed variance compared to IC,
in particular for TREES-reg (Fig. 6). The combination of both tree-ring and ice core proxies (TIC-reg and TIC-MDN) provides
the best reconstruction in the South-Western Andes, in terms of correlation coefficients and of reconstructed variance.

Only one proxy record is used in both TREES-reg and TREES-MDN in South America (i.e, SAm_11). For this site (i.e.,
the grid cell closest to the proxy site), the correlation coefficients are similar for all the experiments, so that the different proxy
records all bring some information, but when tree-ring records are included, an improvement of the reconstructed variance
is obtained. More generally, at all spatial scales in South America the performance of TREES-reg in terms of correlation
coefficients and reconstructed variance is slightly better than TREES-MDN.

**Table 4.** Pearson correlation coefficient between the reconstructed and GMF near-surface air temperature or precipitation over 1901–2000
CE for different DA experiments (Table 1, with ice core proxies only, with tree-ring proxies only or with both) in different regions (Sect. 3)
and proxy sites (Fig. 1). Standard deviation of the reconstruction in °C (near-surface air temperature) or in mm year$^{-1}$ (precipitation). *SA*
stands for South America; *SAmSW* for the South-Western border of South America in the Andes where the TRW proxy sites are located; *NZ*
for New Zealand. *Aus* refers to a site in New Zealand and *SAm*, in South America. Asterisks stand for significant correlation coefficients at
the 95% confidence level.

| | Correlation | | | | |
|---|---|---|---|---|---|
| | TIC-reg | TIC-MDN | IC | TREES-reg | TREES-MDN |
| **Precipitation** | | | | | |
| SA | 0.12 | -0.05 | 0.28* | 0.02 | -0.20* |
| SAmSW | 0.37* | 0.35* | 0.34* | 0.33* | 0.26* |
| SAm_11 | 0.26* | 0.22* | 0.22* | 0.22* | 0.14 |
| **Near-surface air temperature** | | | | | |
| NZ | 0.55* | 0.54* | 0.50* | 0.53* | 0.52* |
| Aus_005 | 0.20* | 0.26* | 0.29* | 0.19 | 0.24* |
| Aus_030 | 0.49* | 0.41* | 0.30* | 0.57* | 0.48* |

| | Standard deviation | | | | |
|---|---|---|---|---|---|
| | TIC-reg | TIC-MDN | IC | TREES-reg | TREES-MDN |
| **Precipitation** | | | | | |
| SA | 17.82 | 15.29 | 8.31 | 17.22 | 18.53 |
| SAmSW | 63.59 | 51.79 | 33.79 | 52.00 | 38.50 |
| SAm_11 | 59.51 | 41.84 | 17.63 | 57.74 | 41.38 |
| **Near-surface air temperature** | | | | | |
| NZ | 0.19 | 0.15 | 0.11 | 0.18 | 0.13 |
| Aus_005 | 0.21 | 0.17 | 0.11 | 0.21 | 0.16 |
| Aus_030 | 0.17 | 0.13 | 0.07 | 0.18 | 0.13 |



**Figure 6.** Reconstructed (in color) and GMF (in black) temperature (in °C; relative to 1901–2000 CE) or precipitation anomalies (in mm year$^{-1}$; relative to 1901–2000 CE) over 1901–2000 CE for different DA experiments (Table 1, with ice core proxies only, with tree-ring proxies only or with both) in different regions (Sect. 3) and proxy sites (Fig. 1). *Aus* refers to a site in New Zealand and *SAm*, in South America. The uncertainty associated with each reconstruction is represented by the shaded areas.





In New Zealand, the correlations with observations are similar in all the experiments (NZ; Table 4). In other words, tree-ring proxies only are able to provide some skill at the scale of the New Zealand, even without assimilating ice core proxies, and with a slightly higher reconstructed variance (especially for TREES-reg). The contrast between the reconstructed variance in IC and TREES-reg or TREES-MDN experiments is stronger at the scale of the proxy site (Aus_005 and Aus_030, the tree-ring proxy sites assimilated both with MAIDEN and the regression model). At the Aus_030 proxy site the correlation coefficients are even higher for the experiments assimilating tree-ring proxies only compared to ice core proxies only. However, the reconstructed variance is still underestimated compared to observations (Fig. 6).

**Table 5.** Pearson correlation coefficient between the reconstructed and GMF near-surface air temperature or precipitation over 1901–2000 CE for different DA experiments (Table 1, with tree-ring proxies only, with the full observation error or the half of the observation error; see Sect. 3 for more details) in different regions (Sect. 3) and proxy sites (Fig. 1). Standard deviation of the reconstruction in °C (near-surface air temperature) or in mm year$^{-1}$ (precipitation). *SA* stands for South America; *SAmSW* for the South-Western border of South America in the Andes where the TRW proxy sites are located; *NZ* for New Zealand. *Aus* refers to a site in New Zealand and *SAm*, in South America. Asterisks stand for significant correlation coefficients at the 95% confidence level.

| | Correlation | | | |
|---|---|---|---|---|
| | TREES-reg | TREES-MDN | TREES-reg-05 | TREES-MDN-05 |
| **Precipitation** | | | | |
| SA | 0.02 | -0.20 | -0.05 | -0.19 |
| SAmSW | 0.33* | 0.26* | 0.37* | 0.24* |
| SAm_11 | 0.22* | 0.14 | 0.21* | 0.16 |
| **Near-surface air temperature** | | | | |
| NZ | 0.53* | 0.52* | 0.51* | 0.50* |
| Aus_005 | 0.19 | 0.24* | 0.23* | 0.25* |
| Aus_030 | 0.57* | 0.48* | 0.57* | 0.48* |

| | Standard deviation | | | |
|---|---|---|---|---|
| | TREES-reg | TREES-MDN | TREES-reg-05 | TREES-MDN-05 |
| **Precipitation** | | | | |
| SA | 17.22 | 18.53 | 25.12 | 26.73 |
| SAmSW | 52.00 | 38.50 | 75.51 | 60.58 |
| SAm_11 | 57.74 | 41.38 | 79.44 | 60.13 |
| **Near-surface air temperature** | | | | |
| NZ | 0.18 | 0.13 | 0.23 | 0.21 |
| Aus_005 | 0.21 | 0.16 | 0.27 | 0.26 |
| Aus_030 | 0.18 | 0.13 | 0.22 | 0.22 |

The underestimation of the variance compared to the observations could arise from a too low confidence given to the assimilated tree-ring records in the data assimilation procedure, which is directly related to the error associated with the PSMs (Sect. 2.5). In order to test this hypothesis, we have reduced this error by two. This leads to an increase of the reconstructed variance at all spatial scales in both South America (precipitation) and New Zealand (near-surface air temperature; Table





**Figure 7.** Reconstructed (in color) and GMF (in black) temperature (in °C; relative to 1901–2000 CE) or precipitation anomalies (in mm year$^{-1}$; relative to 1901–2000 CE) over 1901–2000 CE for different DA experiments (Table 1, with tree-ring proxies only, with the full observation error or the half of the observation error; see Sect. 3 for more details) in different regions (Sect. 3) and proxy sites (Fig. 1). *Aus* refers to a site in New Zealand and *SAm*, in South America. The uncertainty associated with each reconstruction is represented by the shaded areas.





5 and Fig. 7), and in particular for TREES-MDN in New Zealand. This does not translate, however, into higher correlation coefficients, since they virtually remain unchanged (Table 5). In an ideal experiment where the relationship between climate and tree-ring growth would be perfectly reproduced, we could thus expect the observed variance to be more skillfully reconstructed

by the ensemble mean reconstruction.

More generally, this sensitivity analysis shows that tree-ring proxies have a positive effect on the reconstructed mid-latitude climate changes in our framework with both the regression- or process-based dendroclimatic PSM, especially at the regional to local scale. The fact that the reconstructed variance can increase at the more local scale when assimilating tree-ring proxies alone indicates that the number of tree-ring proxies assimilated in our experiments likely has an impact on the reconstructed

variance. If more proxies were available, we could expect that this local increase in the variance would lead to an increase at larger spatial scale. The definition of the error related to the PSMs likely plays a role as well. In particular, the computation of the error for MAIDEN is stricter than for the regression model (i.e., computed on the verification period and on the calibration period, respectively; Sect. 2.5) and the number of proxy records assimilated with MAIDEN is lower as well, which may advantage the regression-based PSM.

## 4.3 Data assimilation based reconstruction of the Southern Hemisphere climate over the last 400 years

In this section, we compare the DA-based reconstructions with the process- or regression-based PSM over the past four centuries, to expand to longer timescales the conclusions obtained from the comparison of the two methodologies over the 20th century. If we first look at the regional reconstructions of near-surface air temperature (Fig. 8) and precipitation (Fig. 9) for the four regions, we observe that the ensemble spread is in general large, contrasting with an interannual variability of the

ensemble mean reconstructions that is generally small. This could indicate that we overestimate the error associated with the PSM, and therefore underestimate the confidence of TRW data in the DA procedure. Besides, as highlighted before, this is at least partly due to the small number of assimilated TRW proxies in the mid-latitudes that prevents the prior distribution given by the climate model to be effectively constrained by the available observations. This pattern is stronger when the spatial scale is large (e.g., South America), and weaker when the spatial scale is small (e.g., Tasmania) and closer to the proxy sites. This is

particularly true for MAIDEN, for which we assimilated half the number of TRW proxy records compared with the regression model.

In New Zealand, both TIC-reg and TIC-MDN display a smaller spread and larger variability of the reconstructed near-surface air temperature compared to the other regions (Fig. 8), which suggests that the DA has been more effective in updating the climate model simulations with the available proxy records. The reconstructed near-surface air temperature in New Zealand

was also well evaluated against GMF observations over 1901–2000 CE in Sect. 4.1 (Table 3). The low-frequency variability of the two reconstructions over the last 400 years is relatively similar overall, with a very small trend before the industrial era (until around 1850-1900 CE) followed by an increase in temperature. TIC-reg and TIC-MDN display a significant trend of 0.047 and 0.023°C per decade during the 20th century (against 0.062°C per decade in the GMF observations) and a significant trend of 0.001 and non-significant trend of 0°C per decade before 1900 CE, respectively. The correlation coefficient between both

reconstructions is 0.63 (significant at the 95% confidence level), which indicates that the trend and timing of the reconstructions





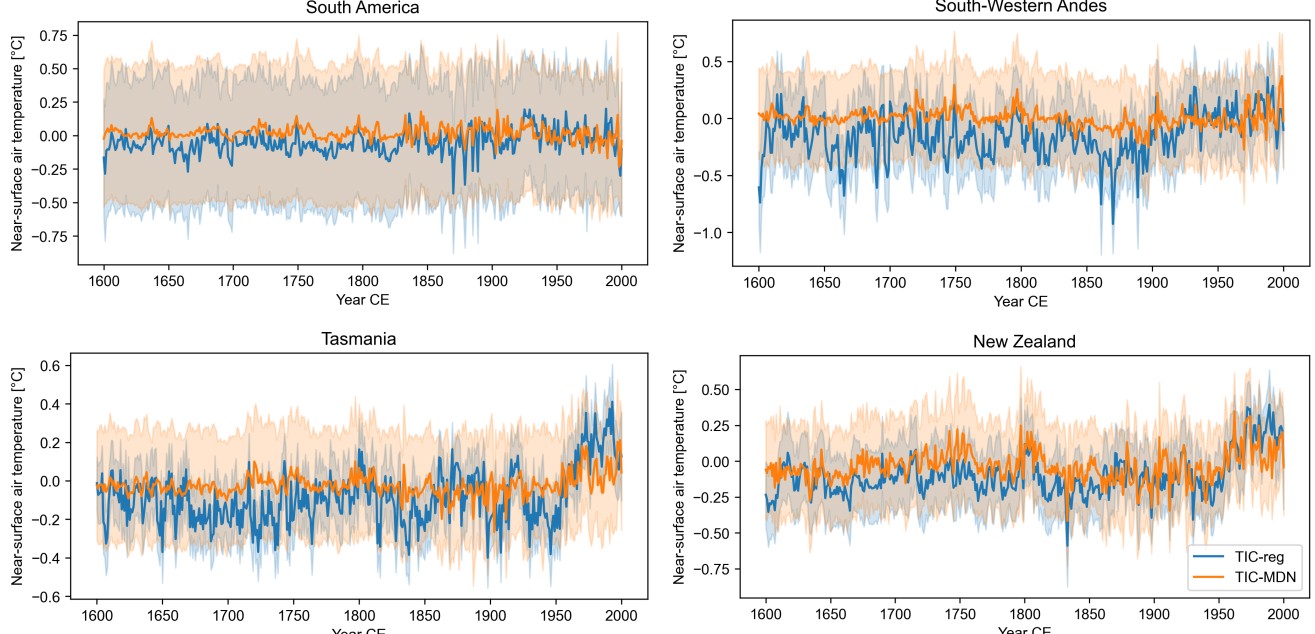

**Figure 8.** Reconstructed TIC-reg (in blue) and TIC-MDN (in orange) near-surface air temperature anomalies (in °C; relative to 1901–2000 CE) over 1600–2000 CE (Table 1) in different regions (Sect. 3). The uncertainty associated with each reconstruction is represented by the shaded areas.

are close to each other. However, TIC-MDN displays higher near-surface air temperature anomalies before 1850 CE than TIC-reg (-0.05°C in average for TIC-MDN, against -0.14°C for TIC-reg, relative to 1901-2000 CE). In Tasmania, there are no assimilated sites for MAIDEN (Figs. S2 and S6) and four assimilated sites for the regression over a very small area (Figs. S3 and S9), which may explain the difference between the reconstructed near-surface air temperature of TIC-reg and TIC-MDN (Fig. 8). This is consistent with the results obtained over the 20th century. Finally, we observe a difference in the reconstructed low-frequency variability between TIC-reg and TIC-MDN. Compared with TIC-reg, TIC-MDN tends to reconstruct a weaker temperature difference between the preindustrial and industrial era, particularly in South America, South-Western Andes and Tasmania.

For precipitation (Fig. 9), the South-Western Andes is the region for which the ensemble spread is the smallest and the interannual variability the largest for both TIC-reg and TIC-MDN. Both experiments are also skillful for reconstructing precipitation in this region when compared with the state-of-the-art climate datasets over the last century (Table 3; Sect. 4.1). The correlation coefficient between both reconstructions is 0.58 and is statistically significant, so that the trend and timing of the reconstructions are close to each other. Both reconstructions display a very small trend before 1900 CE (a significant trend of -0.586 and a non-significant trend of -0.042 mm year$^{-1}$ per decade for TIC-reg and TIC-MDN, respectively) and a negative trend after 1900 CE (a significant trend of -12.79 and -8.89 mm year$^{-1}$ per decade for TIC-reg and TIC-MDN, respectively;





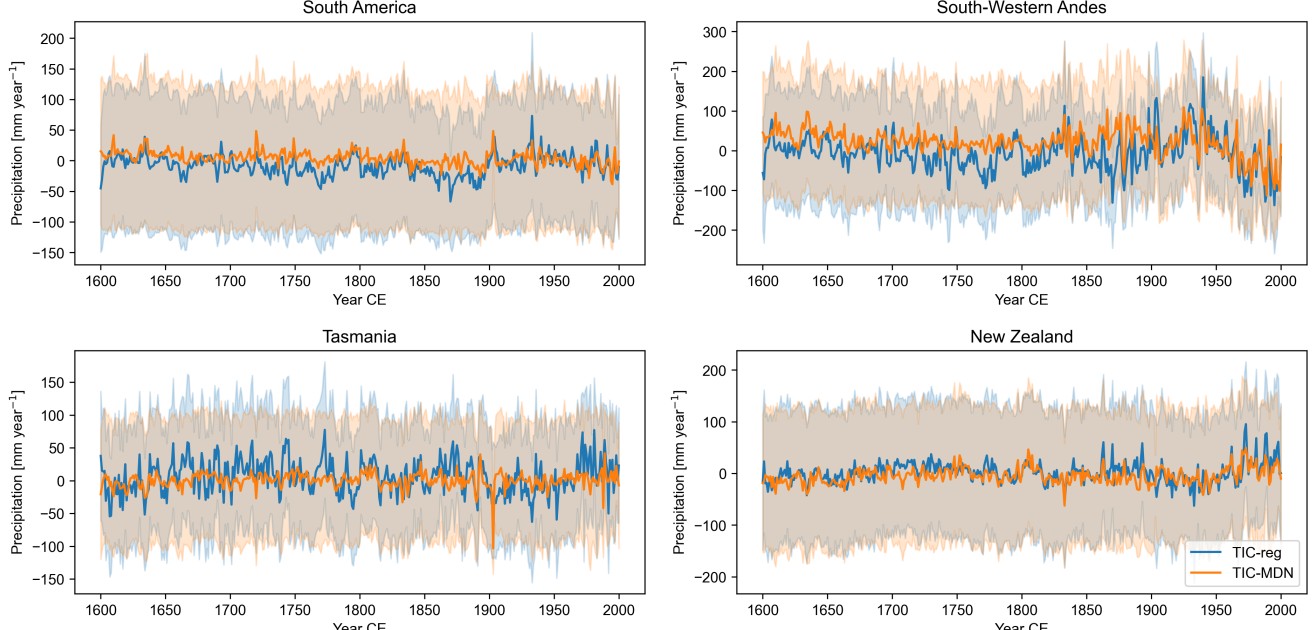

**Figure 9.** Reconstructed TIC-reg (in blue) and TIC-MDN (in orange) precipitation anomalies (in mm year$^{-1}$; relative to 1901–2000 CE) over 1600–2000 CE (Table 1) in different regions (Sect. 3). The uncertainty associated with each reconstruction is represented by the shaded areas.

against a non-significant trend of -1.26 mm year$^{-1}$ per decade in the GMF observations). However, TIC-MDN displays higher anomalies before 1900 CE than TIC-reg (28.23 mm year$^{-1}$ in average for TIC-MDN against -5.70 mm year$^{-1}$ for TIC-reg, relative to 1901-2000 CE).

At the regional scale, we have highlighted from the reconstructed near-surface air temperature and precipitation that the experiments with the statistical or process-based PSM show different amplitudes of climate change over the past 400 years, even if close in terms of correlation (i.e, the trend and timing of change). It is not possible, of course, to determine which one is the most accurate. This difference in the reconstructions could be related to the number of proxies included in each of the DA experiments, and their associated error (Sect. 2.5). This also points out that the commonly used linear regression may also not be fully able to always reflect properly the full spectrum of the climate variability over the past centuries. Alternatively, MAIDEN

may underestimate low-frequency trends, for instance because it overestimates the effect of $CO_2$ concentration on tree growth (Rezsöhazy et al., 2021) which leads to an underestimation of the near-surface temperature low-frequency variability. It may also be related to the calibration of MAIDEN that may not be able to properly capture the low-frequency signal in tree-ring series.

Finally, we observe an increase of the reconstructed variability in the precipitation of the South-Western Andes (Fig. 9),

particularly for TIC-MDN, from around 1800–1850 CE, i.e., when most of the ice core records start to become available (Fig.





S11). Before 1830 CE, the standard deviation of the reconstruction is 30.7 and 21.0 mm year$^{-1}$ and, after 1830 CE, 56.3 and 44.2 mm year$^{-1}$ for TIC-reg and TIC-MDN, respectively.

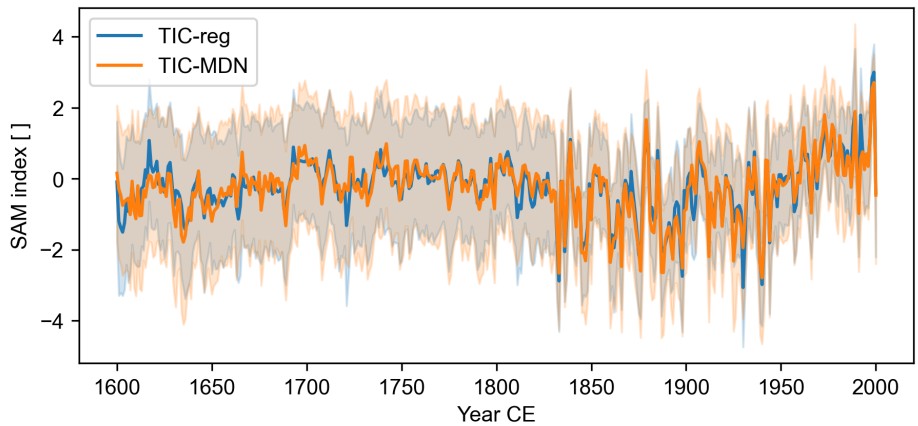

**Figure 10.** Z-scored (relative to 1901–2000 CE) TIC-reg (in blue) and TIC-MDN (in orange) reconstructed SAM index over 1600–2000 CE (Table 1). The uncertainty associated with each reconstruction is represented by the shaded areas.

This increase of interannual variability is even more striking in the reconstructed SAM index for both reconstructions (Fig. 10). Before around 1830 CE particularly, too few TRW proxy sites have been assimilated in the mid-latitudes to be able to

reconstruct the large-scale variability adequately (Fig. S11). The standard deviation of the z-scored reconstructed SAM index is 0.52 and 0.51 before 1830 CE and 1.05 and 1.11 after 1830 CE for TIC-reg and TIC-MDN, respectively. However, the SAM reconstructions are still highly and significantly correlated with each other ($r = 0.91$), and look essentially the same over the whole 1600-2000 CE period (Fig. 10). Before 1830 CE, when the number of ice core records is more limited (Fig. S11), the reconstructions are less correlated ($r = 0.77$; and $r = 0.95$ after 1830 CE).

Lastly, over the last 400 years, the MAIDEN model generally reproduces the observed tree growth well in terms of correlation coefficients at the six assimilated sites (with a mean correlation of 0.53; Fig. S12). However, as previously highlighted in Sect. 4.1, the simulated variance is largely underestimated compared to the observations, which is not surprising regarding the reconstructions of near-surface air temperature and precipitation of the past 400 years.

## 5   Conclusions

In this study, we have included for the first time a dendroclimatic process-based model into a data assimilation procedure to reconstruct the past climate variability of the Southern Hemisphere. All the results from our DA experiments have been compared with the commonly used regression-based PSM. First, we have shown that including the ecophysiological model MAIDEN as a proxy system model into a data assimilation procedure is technically possible. The assimilation of actual



TRW records with a tree-growth mechanistic model has never been published before. Knowing that it is at least achievable is
promising.

The general pattern of near-surface air temperature, precipitation and the atmospheric circulation of the mid-to-high latitudes
of the SH is well reproduced by the DA experiment with MAIDEN over the recent period. Over the last century, the DA
experiment with MAIDEN is also able to produce reconstructions that are well correlated with different climate fields, such as
near-surface air temperature, precipitation and atmospheric circulation in some regions of the Southern Hemisphere, with only
a few TRW proxy records assimilated. The DA-based reconstructions with the regression have a similar skill as with MAIDEN.

Over the instrumental era, the skill of MAIDEN as a PSM in the present DA framework is not better but it is at least as good
as the regression-based PSM. Given the advantages that a process-based PSM can still bring beyond the instrumental period in
terms of non-linearity of the relationship between climate and tree growth, but also in terms of climate variables included as
inputs of the model (e.g. atmospheric $CO_2$ concentration), having a comparable skill to a regression-based PSM is encouraging.
There are still opportunities for improvement with the MAIDEN model while for the regression, the perspectives are more
limited. For instance, we could improve the model-data comparison by using an observed tree growth index that is closer to
the variable simulated by MAIDEN, i.e., forest carbon accumulation, by combining density and ring width measurements.
Different shortcomings related to the calibration of the MAIDEN model have also been addressed in Rezsöhazy et al. (2020)
and Rezsöhazy et al. (2021). If solved, this may likely help to increase the number of proxy sites where MAIDEN could be
validated. It could also improve the performance of MAIDEN in simulating the relationship between climate and tree growth,
and so enhance the skill of the DA-based reconstruction using MAIDEN as a PSM. In particular, the calibration procedure could
be more informative in the future, for instance in the definition of the prior ranges of the calibrated parameters, that could be
set by species or biome. Different data sources, such as satellite data and the associated vegetation indexes, could also be used
to better inform the different biological processes in the model (e.g. related to photosynthesis or phenology). Moreover, in this
study, the criteria for including a TRW proxy record with the regression had to be less strict than for MAIDEN. MAIDEN was
indeed validated at all the assimilated sites while the regression was not. This may provide a better stability of the MAIDEN
model.

It appears clearly that our DA framework does not include enough TRW proxy records in the mid-latitudes to properly
reproduce the observed variance of near-surface air temperature and precipitation with both TRW PSMs, particularly at larger
spatial scale (e.g., South America). The Southern Hemisphere counts indeed few TRW proxy records (PAGES 2k Consortium,
2017) but offers an interesting framework as few reconstructions of the Southern Hemisphere climate exist to date (Neukom
et al., 2011, 2014). This pattern of variance becomes even more striking before 1830 CE, where we observe a smaller interan-
nual variability of the reconstructed near-surface air temperature and precipitation in some regions of the mid-latitudes, and of
the reconstructed SAM index, compared to the last 150 years. However, the timing of the climate changes is well constrained
in some localized areas of the mid-latitudes, e.g., the South-Western Andes, New Zealand and Tasmania, as shown by the
significant and relatively high correlations with the observations. As a consequence, the MAIDEN model simulates tree-ring
growth series that are generally well correlated with the observations over the last 400 years, but which display a small variance
compared to the observations.



Despite this, at the regional scale, we have highlighted a difference between the two DA experiments in the reconstructed amplitude of the climate change in the preindustrial era relative to the 20th century. Using MAIDEN as a PSM for TRW records tends to reconstruct an average warmer and wetter climate than the regression-based PSM. This questions the almost exclusive use of linear statistical PSMs for tree-ring proxies in data assimilation to reconstruct the past climate changes. However, this would need an in-depth investigation with more TRW proxy records in order to be confirmed, as well as additional analysis to ensure the validity and robustness of MAIDEN and its calibration in our framework, and more generally of sophisticated tree-growth PSMs.

For now, our framework only offers a limited skill of the reconstructions, and we are far from providing a complete hemispheric reconstruction of atmospheric fields in the Southern Hemisphere. However, our results are encouraging for MAIDEN, and more generally for the use of dendroclimatic process-based models in DA. In the future, mechanistic models like MAIDEN will likely become an important tool of the DA framework that would help overcome the potential bias associated with regression models. However, this would require to continue to improve those models, including their calibration, so that they could be applied more efficiently and more skillfully at large spatial scale. Calibrating a model like MAIDEN at large spatial scale remains a challenge, and incorporating it into a data assimilation procedure is still more technically time consuming than the traditional statistical approach. Finally, more investigation of the performance of the MAIDEN model as a PSM in the DA procedure compared with the commonly used regression model would also be needed using more TRW proxy records in order to fully document the added value of using a dendroclimatic process-based model in the data assimilation framework.

*Competing interests.* The authors declare that there is no conflict of interest.

*Acknowledgements.* JR was F.R.S-FNRS research fellow, Belgium (grant no. 1.A841.18); QD is F.R.S-FNRS postdoctoral researcher, Belgium; HG is research director at F.R.S.-FNRS, Belgium; JG is emeritus research director at CNRS, France. This research has been supported by the Belgian Research Action through Interdisciplinary Networks (BRAIN.be) from the Belgian Science Policy Office in the framework of the "East Antarctic surface mass balance in the Anthropocene: observations and multiscale modelling (Mass2Ant)" project (contract no. BR/165/A2/Mass2Ant). Computational resources have been provided by the supercomputing facilities of the Université catholique de Louvain (CISM/UCL) and the Consortium des Équipements de Calcul Intensif en Fédération Wallonie Bruxelles (CÉCI) funded by the Fond de la Recherche Scientifique de Belgique (F.R.S.-FNRS) under convention 2.5020.11 and by the Walloon Region.



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
