# Peer review of "Using a process-based dendroclimatic proxy system model in a data assimilation framework: a test case in the Southern Hemisphere over the past centuries"

_Climate of the Past, 2021_

## Referee Comment (RC2)

Review of

'Using a process-based dendroclimatic proxy-system model in a data assimilation framework: a test case in the Southern Hemisphere over the past centuries.'

by J. Reszöhazy, Q. Dalaiden, F. Klein, H. Goosse, and J. Guiot

**Recommendation: minor revisions**

This manuscript evaluates whether using a process-based dendroclimatological proxy system model in the context of paleoclimate data assimilation provides better results than using simple regression-based tree growth models. This is a relevant research question which has not been addressed yet. For the specific case study presented in the manuscript it is found that the performance of the two methods is similar, with some differences depending on regions and performance measures. The analysis is technically sound and the study is a valuable contribution to the further improvement of data assimilation methods. However, there are some general points and some details that could have been discussed more clearly. I recommend publication after the specific points listed below have been addressed.

**Specific comments**

1)
Lines 25-28, 36-46: The problem with statistical models are not only assumptions on linearity and stationarity, but also that they are 'inverse models'. It should be made clear that multiple climate states may lead to the same response in proxy data, and that this can be taken into account with PSMs, whereas inverse models assume invertibility of the relationship, which may not be the case. It should also be mentioned that the PSMs are a specific form of the 'observation operator' in the general DA framework.

2)
Lines 30-31: 'impact' should be replaced by 'contribution', and 'natural' with 'natural, random' or 'natural, internal', because 'impact' describes an external influence on a system, and 'natural' variability includes natural forced variability.

3)
Lines 46-50: The setup of the pseudoproxy studies and of the role of VS-Lite in them should be better explained, so that the main aspects become clear without reading the references.

4)
Lines 56-58: A very brief explanation of the setup of the calibration and validation of the MAIDEN model in Reszöhazy et al. (2020, 2021) would be good. For instance, what are the inputs and outputs?

5)
Lines 62-64: Has MAIDEN not been evaluated in the Northern Hemisphere or did it not perform well?

6)
Lines 66-68: This is the first time oxygen isotopes data are mentioned. It is explained later that these are available from the isotope-enabled GCM simulations, but it would be good to briefly mention this already here.

It is unclear why the different types of proxies are linked to different spatial scales. Is the argument that by using only TRW data large areas would not be covered at all with proxy data, or that isotope and/or snow cover data are linked to larger spatial scales than TRW data. If it is the latter, why is this case?

7)
Lines 116-118: The discussion of dynamical consistency should be more precise and avoid overselling.

The individual particles for a given timestep are dynamically consistent climate states, but in an offline DA there is no dynamical consistency in time. This is not a problem if the timesteps are so long that the atmospheric states are almost independent. However, the dynamics of the ocean and cryosphere components of the climate system involves also very long timescales, and the ocean and cryosphere states influence the atmospheric states.

Moreover, dynamical consistency in space and between variables involves non-linear equations and the weighted ensemble mean is therefore not dynamically consistent.

8)
Line 146: 'anomalies ... are subtracted from the TRW timeseries' seems wrong. Please rephrase.

9)
Line 153: Genitive s after reference should not be there.

10)
This is the first time the 'observation operator' is mentioned. The fact that PSMs are one example for observation operators in DA should have been mentioned already in the introduction (see also comment 1).

11)
Lines 174-176: As far as I understand the annual quantity of carbon that is added to a tree (Dstem) is proportional to the added tree volume. i.e. proportional to r * delta_r, with r the radius of the tree and delta_r the tree ring width. It seems therefore problematic to compare Dstem only with delta_r. It would be good to add a comment on this in the text.

12)
Line 186: A correlation of 0.3 means that less than 10% of the variance is explained. A comment on how this may affect the DA would be helpful.

13)
Line 225: The standard meaning of 'validation' is quantification of skill, not demonstration that the skill is high. The statement should be replaced by something like 'only two locations satisfied the same selection criterion as MAIDEN'.

14)
Lines 310-312: The meaning of CE and the reason for using it should be better explained. For a linear prediction model the correlation includes the complete information about the amplitude of the predictions, because the squared correlation is the explained variance. The reason this is not sufficient in the context of the study is that no linear models are applied to correct the simulated variables. The CE compares the variance of the residuals with the variance of the predictand, with $CE = 1$ associated with perfect predictions, $CE = 0$ with a residual variance identical to predicting the observed mean, and $CE < 0$ with larger residual variance than when predicting the mean.

15)
The underestimation of variance in the reconstructions is discussed in several places and attributed to weak constraints on the prior through the proxy data, and thus similar weights for a large number of particles. This is in principle correct. However, this is a fundamental issue with the particle filter, and potentially with other data assimilation methods, and not all relevant aspects become clear. The paper would benefit from a more systematic discussion of the reasons for variance underestimation, including at least the following points:

- Individual members of ensemble climate simulations or sequences of selected timesteps from individual members have in the ideal case realistic temporal variability on all spatial scales.

- Any averaging of random, non-forced variability will reduce variability. This means that any ensemble mean will always have unrealistically low variability, regardless of how it is constructed. However, the extent of the reduction depends on whether members with more similar or more different variability are used for calculating the weighted ensemble mean.

  In contrast to unweighted ensemble means the Particle Filter gives high weights to ensemble members that match the observations. If the empirical information strongly constrains the particle selection, the particles will be more similar than in a less constrained case, and there will be less underestimation of variance.

- The similarity measure is determined at the locations of the proxies, but the weights given to each particle are independent of location (more detail on this in the manuscript would be helpful). Particles that have similar states at the proxy locations don't necessarily have similar states at other locations, and the reduction of variance in unconstrained locations is therefore likely to be larger than in constrained locations.

16)
Lines 351, 394: replace 'constrain' with 'constraint'.

17)
Line 354-359: These statements are very unclear. A TRW PSM does not assess 'errors in the simulated variance of climate signals'. Please clarify the argument.

18)
Line 361: 'fully verify our DA procedure' is not well phrased.

19)
Lines 362, 396: 'if' should be replaced with 'whether'.

20)
Lines 386-387: The comment on the potential influence of the different number of tree ring records for reg and MAIDEN is helpful, but this potential issue should be already mentioned in the introduction and/or method section, and the experimental setup using different numbers should be justified.

---

## Author Comment (AC1)

Dear editor and reviewers, we would like first to thank you for your useful feedback and comments on our manuscript. You can find here below the Referee's comments in *italics* and our answers in blue. In **bold**, you can find the modifications that will be made to the manuscript.

**Referee#1**

**General comments**

*The manuscript "Using a process-based dendroclimatic proxy system model in a data assimilation framework: a test case in the Southern Hemisphere over the past centuries" by Rezsöhazy et al. is a good example of using a process-based model MAIDEN as new tool in the assimilation-based climate (particularly, near-surface air temperature, precipitation and winds ) reconstructions in the Southern Hemisphere over the last 400 years. I note that the Southern Hemisphere is poor investigated area in the context of climate reconstructions so far. Mostly the published long-term climate reconstructions are based on different regression approaches to make the link between the GCMs outputs and the proxy observations obtained from tree rings, ice cores, etc. In this work the authors made a considerable effort to use the process-based MAIDEN (that removes a number of limitations inherent in conventional regressions) as a proxy system model in a data assimilation procedure, using as a test the reconstructions of three climatic variables.*

*The paper is well structured and written. The introduction provided a comprehensive overview of the background information and the pertinent literature, and it demonstrated the need for the current study. They involved a wide range of statistical and modeling techniques, included original ideas, as well as several international global databases for testing their hypotheses and confirming their results.*

We would like to thank the Referee for this positive general feedback, for the careful evaluation of our manuscript as well as for the useful comments that will be addressed in the revised version as specified here below.

*But there is an issue which can be considered in the MS. The particle filter method described in Sect. 2.1 is not used often in research. Could the authors describe in details how were those climate particles used to adjust different climate reconstructions used as inputs of MAIDEN over the past 400 years (see the Section 2.4.1)? Could the authors produce some visualization (figure) how does this algorithm work?*

Previous studies have demonstrated that the particle filter is a valid method for offline data assimilation (e.g., Widmann et al., 2010; Dubinkina et al., 2011; Goosse et al., 2012; Klein and Goosse, 2018; Dalaiden et al., 2021). In particular, one of the advantages of this method compared to other methods is that it respects the climate system dynamics as represented in the climate model because only the weight of the sampled particles are modified by comparison with the proxy data during the assimilation. Additionally, it does not involve any linearization or inversion of the PSM. The method does not make any assumptions on the prior and posterior distributions of the state of the climate system, and it can be used without an a priori knowledge of the model.

The hypotheses that underline other filters such as the Kalman filter can be problematic when using a process-based model as MAIDEN for assimilating tree-ring series, especially

regarding the assumption of a Gaussian distribution of the tree-ring simulations (see the work of Steiger and Smerdon, 2017 with VS-Lite). This is one of the reasons to use the particle filter here.

A more in-depth explanation of how the particle filter works can be found in my PhD thesis (freely available here: https://dial.uclouvain.be/pr/boreal/en/object/boreal%3A258209/datastreams), in the Appendix D.1 from p.159. This will be referred to as follows on l119, at the end of Section 2.1 : "**More details on the implementation of the particle filter in our experiments can be found in Rezsöhazy (2021).**"

In particular, the following illustration provides  a schematic representation of the procedure of offline data assimilation with a particle filter:

[Figure]

[Figure]

Before the assimilation, all the N (= 8 in our example) particles (blue circles) sampled in the model states (vertical axis) have the same weight, which is proportional to the size of the circle, whatever the year considered (t and t+1 in the example) because the prior is fixed throughout the assimilation procedure. The distribution of the available observations with uncertainties for each year is represented by the red curve. After the assimilation, the particles have been weighted relatively to their associated likelihood, given the available observations for each year, and the size of the circle is modified accordingly. When the weight of a particle is zero, it does not appear anymore on the figure.

*Is there some public domain where the code of the algorithm is located?* The code is freely available upon request.

*The principal idea of the MS is to use the certain process-based model as a proxy system model. What is a reason to use an additional proxy data (e.g. delta 18O) which was used in data assimilation procedure tradionally (through the linear regression)?*

The idea behind the assimilation of other proxy data is to cover a vast area of the Southern Hemisphere (i.e., Antarctica), where no tree-ring proxy data can be found, in order to provide a comprehensive reconstruction of the Southern Hemisphere climate. By doing so, in addition to reconstructing the local climate where tree-ring proxy data are available, we also reconstruct the large-scale climate. The isotopic content from Antarctic ice cores (i.e., $\delta^{18}O$) is directly simulated in the climate model, so that no proxy system model is used. For the SMB, a simple assumption (i.e., the difference between precipitation and sublimation/evaporation (P– E) from iCESM1 is comparable to the snow accumulation observations) is made to assimilate the observations, and has been shown to be valuable particularly at large spatial scale (see Agosta et al., 2019; Souverijns et al., 2018; van Wessem et al., 2018 for instance).

This will be made clearer in the manuscript on l149:"**In addition to tree-ring proxy records which are the focus of this paper through the use of a process-based PSM (Sect. 2.4.1), we also assimilate other proxy data, i.e. ice core records, to cover a vast area of the Southern Hemisphere (i.e, Antarctica) where no tree-ring proxy data can be found, in order to provide a comprehensive reconstruction of the Southern Hemisphere climate.**"

*I would suggest to publish the MS after minor revisions.*

**Specific comments**

*Section 45: The term 'pseudoproxies' should be clarified.*

The following sentence will be added on l48: "However, so far, they have never been used in a DA procedure with actual tree-ring proxy data, but only with pseudoproxies (**i.e., synthetic proxy data**; Dee et al., 2016; Acevedo et al., 2017; Steiger and Smerdon, 2017). **In these studies, the VS-Lite model (Tolwinski-Ward et al., 2011) is forced by climate model outputs in order to provide synthetic tree-ring proxy data. These synthetic proxy data are then incorporated in a DA framework, providing a controlled environment to evaluate the DA procedure and resulting reconstruction skill. In those experiments, using the non-linear VS-Lite model as a PSM for tree-ring data has the potential to**

**improve** the quality of the reconstruction **compared to the traditional statistical linear approach of tree-ring models.**"

*Section 50: I would suggest to remove 'offline' because this term is introduced later in the MS.* Done.

*Section 75: The term 'June-Jule year' should be clarified because this is the first mention of it in the work.*

The following words will be added to the manuscript on l77: "To match the seasonality of tree growth in the Southern Hemisphere, all the analysis in this study are performed on a July-June year **(e.g the year 1992 corresponds to July 1992-June 1993), instead of the usual January-December year, suited for trees growing in the Northern Hemisphere.**"

*Section 80: A reference on the 'Bayes theorem' interpretation is needed!*

The reference to van Leeuwen et al. (2009) will be added on l82: "More specifically, the DA procedure is based on the Bayes theorem **(van Leeuwen, 2009)**: (...)"

*Section 90: The statement "the predictability of the variables of interest is smaller than the temporal resolution of DA (i.e, one year in our study)" should be clarified with the corresponding reference.*

The following reference "**Okazaki et al. (2021)**" has been added.

*Section 105: Is the code of the particle filter approach available in some public depository?*

It is available upon request. This will be stated accordingly in the manuscript.

*Section 105: What does 'chosen frequency' mean?*

This means that we can decide the frequency at which we sample a particle/year/model state in a model simulation. In our case, the frequency is one year, so that we take all the years in the model simulation as particles for the data assimilation. Therefore, we increase the probability to match the assimilated observations, which overall results in a better reconstruction. The manuscript will be modified as follows on l107: "**For instance, for a chosen frequency of one year, we sample a particle or model state in the model simulation every one year.**"

*Section 125: The time span used in the work is the last 400 years. What was a reason to mention 'July 850 – June 2005 CE time period' as a time span 'used in this study'?*

This corresponds to the years of the climate model used as particles for the assimilation, but not to the reconstructed time span. For each year of the reconstruction over the past 400 years, all the years of the climate model (July 850- June 2005 CE) are used as particles in the data assimilation framework to compare with the proxy observations, corresponding to a total of 3465 particles (3 members * 1155 "July-June" years) . This will be modified in the revised manuscript to make a clear distinction between the time span of the simulations and of the reconstructions, as follows on l128:"**This corresponds to the time span of the**

**simulations used to sample the particles during the data assimilation procedure, by contrast with the time span of the reconstructions (i.e., the past four centuries).**".

*Section 145: What was a standardization procedure of TRW data used taking into account a non-climate noise (e.g., age-dependent trends, etc) in raw TRW measurements?*

Here, we use the already standardized tree-ring series directly derived from the PAGES2k compilation without any modification. As the tree-ring series are from different studies (referred in the PAGES2k database metadata), the standardization technique used can differ from one tree-ring series to another.

*Section 230: Why did the authors consider positive correlations only? Are the significant negative correlations worse?*

Following Cook et al. (2013), and as stated in PAGES2k Consortium (2017), when tree growth increases under increasing temperature, they are "more likely to produce a reliable expression of past temperature variability compared to trees that respond inversely to temperature, for which the proximal control on growth is moisture stress". The same assumption has been made for precipitation.

Our decision is thus made on the assumption that a tree should grow better under increasing temperature (under temperature-limiting conditions) and precipitation (under moisture-limiting conditions).

The manuscript will be modified as follows on l232:"**Following Cook et al. (2013), and as stated in PAGES2k Consortium (2017), trees growing better under increasing temperature should be more reliable recorders of past temperature variability, because they respond to temperature-limiting conditions. The same assumption can hold for precipitation.**"

*Figure 2. The green dots should be highlighted stronger.* Done.

*Figure 3. What does 'uncertainty' mean? This term should be clarified.* This corresponds to the so-called "ensemble spread" as defined in Sect. 2.1. This has been modified accordingly: "**(i.e., the ensemble spread; Sect. 2.1)**"

*Section 370: What do 'Aus_002 and SAm_24' mean? Clarification is needed.* Those are the names of the tree-ring sites as in the PAGES2k database. A few words will be added in the manuscript on l371: "**(i.e., the name of the tree-ring sites, as in the PAGES2k database; Aus_\* stands for a Tasmanian or New Zealand site, while SAm_\* stands for a South American site).**"

---

## Author Comment (AC2)

Dear editor and reviewers, we would like first to thank you for your useful feedback and comments on our manuscript. You can find here below the Referee's comments in *italics* and our answers in blue. In **bold**, you can find the modifications that will be made to the manuscript.

*This manuscript evaluates whether using a process-based dendroclimatological proxy system model in the context of paleoclimate data assimilation provides better results than using simple regression-based tree growth models. This is a relevant research question which has not been addressed yet. For the specific case study presented in the manuscript it is found that the performance of the two methods is similar, with some differences depending on regions and performance measures. The analysis is technically sound and the study is a valuable contribution to the further improvement of data assimilation methods. However, there are some general points and some details that could have been discussed more clearly. I recommend publication after the specific points listed below have been addressed.*

We would like to thank the Referee for this general feedback, the careful evaluation of our manuscript and for the interesting comments. They will be accounted for in the revised version of the manuscript as specified here below.

**Specific comments**

*1)*

*Lines 25-28, 36-46: The problem with statistical models are not only assumptions on linearity and stationarity, but also that they are 'inverse models'. It should be made clear that multiple climate states may lead to the same response in proxy data, and that this can be taken into account with PSMs, whereas inverse models assume invertibility of the relationship, which may not be the case.*

*It should also be mentioned that the PSMs are a specific form of the 'observation operator' in the general DA framework.*

The manuscript will be modified as follows:

On l25: "**In this so-called transfer function approach (Fritts et al., 1971), the climate field of interest is expressed as a function of the tree-ring proxy, i.e. in the inverse mode (as opposed to the natural direction where tree growth responds to climate). However, while this assumes an invertibility of the relationship between climate and tree growth, this may not always be the case. Additionally,** the use of linear regression is based on the assumption that the relationship between climate and tree growth is linear and stationary over time."

On l39: "More recently, the use of forward proxy system models (hereafter, PSMs) has emerged in order to directly assimilate the proxy time series. **This gives the opportunity to consider the relationship between the proxy and the climate in the natural direction, as opposed to inverse models, and to take into account the combined influence of multiple climate variables on the proxy data.**"
On l39: "Specifically, the PSMs **– one specific form of the observation operator in the DA framework –** make the link between the outputs of the climate model included in the DA procedure and the assimilated proxy observations (...)"

*2)*

*Lines 30-31: 'impact' should be replaced by 'contribution', and 'natural' with 'natural, random' or 'natural, internal', because 'impact' describes an external influence on a system, and 'natural' variability includes natural forced variability.* Done.

*3)*

*Lines 46-50: The setup of the pseudoproxy studies and of the role of VS-Lite in them should be better explained, so that the main aspects become clear without reading the references.*

This will be added to the manuscript on l48: "However, so far, they have never been used in a DA procedure with actual tree-ring proxy data, but only with pseudoproxies (**i.e., synthetic proxy data**; Dee et al., 2016; Acevedo et al., 2017; Steiger and Smerdon, 2017). **In these studies, the VS-Lite model (Tolwinski-Ward et al., 2011) is forced by climate model outputs in order to provide synthetic tree-ring proxy data. These synthetic proxy data are then incorporated in a DA framework, providing a controlled environment to evaluate the DA procedure and resulting reconstruction skill. In those experiments, using the non-linear VS-Lite model as a PSM for tree-ring data has the potential to improve** the quality of the reconstruction **compared to the traditional statistical linear approach of tree-ring models.**"

*4)*

*Lines 56-58: A very brief explanation of the setup of the calibration and validation of the MAIDEN model in Reszöhazy et al. (2020, 2021) would be good. For instance, what are the inputs and outputs?*

A brief explanation will be added as follows on l57:"In Rezsöhazy et al. (2021), the MAIDEN model has been successfully applied **over the last century** to the PAGES2k TRW database **by comparing the simulated annual quantity of carbon allocated to the stem with the observed tree ring widths (both normalized), using as inputs of the model high-resolution climate data. The model has been calibrated over the second half of the 20th century using the Bayesian procedure with Markov Chain Monte Carlo sampling elaborated in Rezsöhazy et al. (2020), and successfully verified over the first half of the century at 20% of the PAGES2k tree-ring sites. The studies have highlighted the potential of MAIDEN to be used as a PSM for DA-based reconstruction of past climate variability, compared to a simple model like VS-Lite (Tolwinski-Ward et al., 2011).**"

*5)*

*Lines 62-64: Has MAIDEN not been evaluated in the Northern Hemisphere or did it not perform well?* It has been evaluated in the Northern Hemisphere and it performed well but we wanted to focus on the Southern Hemisphere where fewer large-scale reconstructions exist for now.

*6)*

*Lines 66-68: This is the first time oxygen isotopes data are mentioned. It is explained later that these are available from the isotope-enabled GCM simulations, but it would be good to briefly mention this already here.*

The manuscript will be modified as follows on l69: "**While the TRW records require the use of a PSM, the ice core records can be directly compared with the outputs of an isotope-enabled climate model.**"

*It is unclear why the different types of proxies are linked to different spatial scales. Is the argument that by using only TRW data large areas would not be covered at all with proxy data, or that isotope and/or snow cover data are linked to larger spatial scales than TRW data. If it is the latter, why is this case?*

It is the first explanation: by using only TRW data large areas will not be covered by proxy data, in particular the high latitudes of the Southern Hemisphere. In order to provide a more exhaustive reconstruction of the Southern Hemisphere climate, we need to incorporate these high-latitude proxy data as well. By using the ice core records, we do not constrain only the climate at the local-scale where the tree-ring proxy data are available but also the large-scale climate. This allows us to provide a large-scale consistency in the reconstruction, for instance by reconstructing the westerly winds.

The manuscript will be modified as follows on l67: "The DA experiments are based on both TRW and ice core (d$^{18}$O and snow accumulation) proxy data to ensure the consistency of the reconstructed large-scale circulation pattern by **assimilating proxy records at mid and high (i.e., Antarctica) latitudes of the Southern Hemisphere and thus** avoiding to only reconstruct small-scale features."

Also, this will be made clearer in the manuscript on l149:"**In addition to tree-ring proxy records which are the focus of this paper through the use of a process-based PSM (Sect. 2.4.1), we also assimilate other proxy data, i.e. ice core records, to cover a vast area of the Southern Hemisphere (i.e, Antarctica) where no tree-ring proxy data can be found, in order to provide a comprehensive reconstruction of the Southern Hemisphere climate.**"

*7)*

*Lines 116-118: The discussion of dynamical consistency should be more precise and avoid overselling.*

*The individual particles for a given timestep are dynamically consistent climate states, but in an offline DA there is no dynamical consistency in time. This is not a problem if the timesteps are so long that the atmospheric states are almost independent. However, the dynamics of the ocean and cryosphere components of the climate system involves also very long timescales, and the ocean and cryosphere states influence the atmospheric states.*

*Moreover, dynamical consistency in space and between variables involves non-linear equations and the weighted ensemble mean is therefore not dynamically consistent.*

We totally agree with the comment of the Referee, even though it is the most consistent we could get in offline data assimilation. We will adapt our manuscript to avoid overselling the dynamic consistency of the particle filter, by adding the proposed statement on l116:"In this framework, **the individual particles for a given timestep are dynamically consistent climate states. However, it is important to keep in mind that even though atmospheric states are almost independent on an annual timescale (as in this study), the dynamics of the ocean and cryosphere components of the climate system involves very long timescales, and their states also influence the atmospheric variables.** At the end of the DA procedure, the weighted mean of the particles for each year over the assimilation period provides an estimate of the climate variables (i.e., the ensemble mean). **Since dynamical consistency in space and between variables involves non-linear equations, the**

**weighted ensemble mean is therefore not fully dynamically consistent. Finally**, the weighted standard deviation of the particles is used to estimate the range of the reconstruction (i.e., the ensemble spread)."

*8)*

*Line 146: 'anomalies ... are subtracted from the TRW timeseries' seems wrong. Please rephrase.*

Indeed, it will be rephrased as follows: "**the anomalies are computed relative to the 1900-2000 CE time period.**"

*9)*

*Line 153: Genitive s after reference should not be there.* Done.

*10)*

*This is the first time the 'observation operator' is mentioned. The fact that PSMs are one example for observation operators in DA should have been mentioned already in the introduction (see also comment 1).* Done (see comment 1).

*11)*

*Lines 174-176: As far as I understand the annual quantity of carbon that is added to a tree (Dstem) is proportional to the added tree volume. i.e. proportional to r * delta_r, with r the radius of the tree and delta_r the tree ring width. It seems therefore problematic to compare Dstem only with delta_r. It would be good to add a comment on this in the text.*

This important point has been discussed in previous papers (Rezsöhazy et al. 2020, 2021) and a few words will be added in the manuscript, as follows on l176: "**In practice, as previously discussed in Rezsohazy et al. (2020,2021), the comparison of the tree-growth observations with the forest carbon accumulation simulated by MAIDEN implies some limitations. For instance, Gennaretti et al. (2018) computed a wood biomass index directly comparable to what MAIDEN simulates. This involves using both tree-ring width and density (earlywood and latewood fractions) data, while the latter is less broadly available. Finally,** we use a combined version of the model from Gea-Izquierdo et al. (2015) and Gennaretti et al. (2017), developed by Fabio Gennaretti (unpublished)."

*12)*

*Line 186: A correlation of 0.3 means that less than 10% of the variance is explained. A comment on how this may affect the DA would be helpful.*

In principle, data assimilation does not require such selection based on correlation because the confidence we give to each record during the data assimilation procedure is proportional to the observation error we assign to each record. The potential lower correlation of the simulated tree growth with the observations is thus taken into account in the observation error and in the DA. In the LMR project (Hakim et al., 2016; Tardif et al., 2019) for instance, such selection is not made. We have chosen to apply this selection, using a low value of the correlation as threshold since tests have shown that including records with a very low correlation increases the noise and reduces the skill of the reconstruction (Franke et al., 2020).

*13)*

*Line 225: The standard meaning of 'validation' is quantification of skill, not demonstration that the skill is high. The statement should be replaced by something like 'only two locations satisfied the same selection criterion as MAIDEN'.* Done as suggested.

*14)*

*Lines 310-312: The meaning of CE and the reason for using it should be better explained. For a linear prediction model the correlation includes the complete information about the amplitude of the predictions, because the squared correlation is the explained variance. The reason this is not sufficient in the context of the study is that no linear models are applied to correct the simulated variables. The CE compares the variance of the residuals with the variance of the predictand, with CE = 1 associated with perfect predictions, CE = 0 with a residual variance identical to predicting the observed mean, and CE < 0 with larger residual variance than when predicting the mean.*

The manuscript will be modified as suggested by adding the following explanation on l311: "While the Pearson correlation coefficient gives an estimation of the linear fit between the reconstructed and observed values, **the CE compares the variance of the residuals with the variance of the predictand. A CE equal to 1 is associated with perfect predictions, a CE equal to 0, with a residual variance identical to predicting the observed mean, and a negative CE, with larger residual variance than when predicting the mean.**"

*15)*

*The underestimation of variance in the reconstructions is discussed in several places and attributed to weak constraints on the prior through the proxy data, and thus similar weights for a large number of particles. This is in principle correct. However, this is a fundamental issue with the particle filter, and potentially with other data assimilation methods, and not all relevant aspects become clear. The paper would benefit from a more systematic discussion of the reasons for variance underestimation, including at least the following points:*

*- Individual members of ensemble climate simulations or sequences of selected timesteps from individual members have in the ideal case realistic temporal variability on all spatial scales.*

*- Any averaging of random, non-forced variability will reduce variability. This means that any ensemble mean will always have unrealistically low variability, regardless of how it is constructed. However, the extent of the reduction depends on whether members with more similar or more different variability are used for calculating the weighted ensemble mean.*

*In contrast to unweighted ensemble means the Particle Filter gives high weights to ensemble members that match the observations. If the empirical information strongly constrains the particle selection, the particles will be more similar than in a less constrained case, and there will be less underestimation of variance.*

*-The similarity measure is determined at the locations of the proxies, but the weights given to each particle are independent of location (more detail on this in the manuscript would be helpful). Particles that have similar states at the proxy locations don't necessarily have similar states at other locations, and the reduction of variance in unconstrained locations is therefore likely to be larger than in constrained locations.*

We totally agree with the Referee and intended to explain those points  in the submitted version of the manuscript. We will thus make it clearer and more systematic in the Conclusions section of the revised version, by adding the proposed statement on l533:"It appears clearly that our DA framework **does not provide reconstructions that properly reproduce the observed variance of near-surface air temperature and precipitation with both TRW PSMs, particularly at larger spatial scale (e.g., South America), likely due to the low number of TRW proxy records assimilated in the mid-latitudes. The Southern Hemisphere counts indeed few TRW proxy records (PAGES 2k Consortium, 2017) but offers an interesting framework as few reconstructions of the Southern Hemisphere climate exist to date (Neukom et al., 2011, 2014). The underestimation of variance is a fundamental issue with the particle filter, but also potentially with other data assimilation methods (e.g. Tardif et al., 2019). In the ideal case, individual members of ensemble climate simulations have realistic temporal variability on all spatial scales. Any averaging of random, non-forced variability will reduce the variability. In other words, any ensemble mean will always have a too low variability, regardless of how it is constructed. However, the extent of the reduction depends on whether members with more similar or more different variability are used for calculating the weighted ensemble mean. In contrast to unweighted ensemble means, the particle filter will give high weights to ensemble members that match the observations. If the empirical information strongly constrains the particle selection, the particles will be more similar than in a less constrained case, and there will be less underestimation of variance. Besides, the similarity measure is determined at the locations of the proxies, but the weights given to each particle are independent of location. Particles that have similar states at the proxy locations do not necessarily have similar states at other locations, and the reduction of variance in unconstrained locations is therefore likely to be larger than in constrained locations.**

**In our reconstructions**, this pattern of variance becomes even more striking before (...)"

The following words will also be added in the manuscript on l116: "In this way, starting from a prior distribution where all the particles have the same weight, the particle filter produces a posterior distribution where the weights of the particles are redistributed according to the observations, for each year of the reconstruction, **and independent of the location**."

*16)*

*Lines 351, 394: replace 'constrain' with 'constraint'.* Done.

*17)*

*Line 354-359: These statements are very unclear. A TRW PSM does not assess 'errors in the simulated variance of climate signals'. Please clarify the argument.*

It has been replaced by the following: "not properly assessed **when using such TRW PSMs to compare the climate model outputs with the proxy observations.**"

*18)*

*Line 361: 'fully verify our DA procedure' is not well phrased.* This has been deleted.

*19)*

*Lines 362, 396: 'if' should be replaced with 'whether'.* Done.

*20)*

*Lines 386-387: The comment on the potential influence of the different number of tree ring records for reg and MAIDEN is helpful, but this potential issue should be already mentioned in the introduction and/or method section, and the experimental setup using different numbers should be justified.*

The following justification will be added at the end of the method section, on l315:
"**It is important to note at this stage that the number of proxy records, but also the tree-ring proxy records themselves, that are assimilated with the statistical or MAIDEN PSMs are different. The regression-based PSM experiment includes twice the number of tree-ring proxy records compared to the one using MAIDEN. This results from different performance of the PSMs when applied to the same subset of SH tree-ring sites (Sect. 2.4.1 and 2.4.2). When working with actual proxy data, such a situation is difficult to avoid but we must be aware that this may have consequences on the skill of the resulting DA-based reconstructions.**"

---

## Author Response (AR2)

Dear Editor, we would like first to thank you for the careful reading of the revised version of our manuscript and for your comments. We will consider all of them in a revised version of the manuscript, as detailed here below. You can find the comments in *italics* and our answers in blue. In **bold**, you can find the modifications that will be made to the manuscript.

*Dear authors,*

*Thank you for submitting a revised version of your manuscript.*

*I consider that it is broadly ready for final publication in Climate of the Past. However, there are a number of instances where you have provided information in the response to the referees that you have not incorporated into the manuscript. Some of this information would be of benefit to the reader, and so I kindly ask you to consider the following comments.*

*Referee #1, Section 2.1:*

*In the response to the referees, you provide extensive information on the implementation of the particle filter. However, in the manuscript you only provide a reference to a PhD thesis. As a general rule, it is not desirable for an article in a journal to rely on a PhD thesis for the description of the methodology. I note that the referee asks you to provide a figure. I do not consider that you need to do this, but I think it would be helpful if you could add around one paragraph to the manuscript that includes the key details from the information in the response.*

We propose to include the following figure (here below) from Rezsöhazy (2021) in the supplementary material with the associated legend and to remove the reference to the PhD thesis in the manuscript. The supplementary figure is referenced on **l106 and on l116.**

The following sentence will also be added on 116l: "In this way, starting from a prior distribution where all the particles have the same weight, the particle filter produces a posterior distribution where the weights of the particles are redistributed according to the observations, for each year of the reconstruction, and independent of the location. **The method does not make any assumptions on the prior and posterior distributions of the state of the climate system, and it can be used without an a priori knowledge of the model, which is a clear advantage of the particle filter. Additionally,** in this framework, the individual particles for a given timestep are dynamically consistent climate states."

[Figure]

year *t* with observations      year *t+1* with observations

**Before assimilation**

*N* sampled particles in the model states

weighting

year *t* with observations      year *t+1* with observations

**After assimilation**

**Legend:** Before the assimilation, all the N (= 8 in our example) particles (blue circles) sampled in the model states (vertical axis) have the same weight, which is proportional to the size of the circle, whatever the year considered (t and t+1 in the example) because the prior is fixed throughout the assimilation procedure. The distribution of the available observations with uncertainties for each year is represented by the red curve. After the assimilation, the particles have been weighted relatively to their associated likelihood, given the available observations for each year, and the size of the circle is modified accordingly. When the weight of a particle is zero, it does not appear anymore on the figure. Figure and legend from Rezsöhazy (2021).

*Referee #1, source code:*

*The referee makes two separate queries in regard to the availability of the source code. You state in your response that the code is available upon request and that you will include a*

*statement to this effect in the manuscript. I apologise if I am missing it, but I cannot locate any such statement. Also, while not required by Climate of the Past, it is nonetheless encouraged to archive source code in a public repository such as Zenodo.*

This will be stated as follows on l119: "**The source code for the implementation of the particle filter in the offline data assimilation framework of this study is available upon request.**"

*Referee #1, section 145:*

*It would be beneficial to include this information on standardisation. Could you please add 1-2 sentences of text to the manuscript?*

This information on standardization will be added as follows on l145: "**Here, we use the already standardized tree-ring series directly derived from the PAGES2k compilation without any modification. As the tree-ring series are from different studies (referred in the PAGES2k database metadata; PAGES 2k Consortium, 2017), the standardization technique used can differ from one tree-ring series to another.** Finally, the TRW records are normalized (...)."

*Referee #2, line 186:*

*This is useful information that clarifies an aspect of your methodology. Could you please include a summary of it in the manuscript?*

This information will be added as follows on l190: "**Data assimilation does not require in principle such a selection based on correlation because the confidence we give to each record during the DA procedure is proportional to the observation error assigned to each record (Sect. 2.5). The potentially low correlation of some simulated tree growth with the observations is thus taken into account in the observation error and in the DA. In the Last Millennium Reanalysis project (Hakim et al., 2016; Tardif et al., 2019), for instance, such a selection is not made. In this study, we apply a selection, using a low value of the correlation as threshold, since tests have shown that including records with a very low correlation can increase the noise and reduce the skill of the reconstruction (Franke et al., 2020).**"

*Kind regards,*
*Steven Phipps*
*Handling Editor*